# Explainable machine learning for profiling the immunological synapse and functional characterization of therapeutic antibodies

Sayedali Shetab Boushehri [1,2,3,4,8], Katharina Essig[5,8], Nikolaos-Kosmas Chlis[5], Sylvia Herter [6], Marina Bacac[6], Fabian J. Theis [2,3], Elke Glasmacher [7] ✉, Carsten Marr [1,2] ✉ & Fabian Schmich [4] ✉

Therapeutic antibodies are widely used to treat severe diseases. Most of them alter immune cells and act within the immunological synapse; an essential cell-to-cell interaction to direct the humoral immune response. Although many antibody designs are generated and evaluated, a high-throughput tool for systematic antibody characterization and prediction of function is lacking. Here, we introduce the first comprehensive open-source framework, scifAI (single-cell imaging flow cytometry AI), for preprocessing, feature engineering, and explainable, predictive machine learning on imaging flow cytometry (IFC) data. Additionally, we generate the largest publicly available IFC dataset of the human immunological synapse containing over 2.8 million images. Using scifAI, we analyze class frequency and morphological changes under different immune stimulation. T cell cytokine production across multiple donors and therapeutic antibodies is quantitatively predicted in vitro, linking morphological features with function and demonstrating the potential to significantly impact antibody design. scifAI is universally applicable to IFC data. Given its modular architecture, it is straightforward to incorporate into existing workflows and analysis pipelines, e.g., for rapid antibody screening and functional characterization.

The formation of an immunological synapse is the first event of the adaptive immune reaction induced by the interaction of a T cell with its corresponding antigen-presenting cell (APC). This rapidly formed cell-cell interface is initiated by the recognition of peptide-loaded major histocompatibility complexes (MHC) by the T cell receptor (TCR). It involves the rearrangement of actin filaments of the cytoskeleton and the recruitment of signaling, co-stimulatory, co-inhibitory, and adhesion molecules to the nascent synapse[1,2]. This process is crucial to trigger and fine-tune T cell responses and ensure intact immune reactions. Dysfunctional immunological synapse formation has been observed in several immune-related disorders[3-8] and has thus been considered a potential target to stimulate or inhibit immune responses by modulating its assembly or function[9-11]. For instance, various therapeutic antibodies were developed that alter immunological synapse

[1]Institute of AI for Health, Helmholtz Zentrum München, German Research Center for Environmental Health, Neuherberg, Germany. [2]Institute of Computational Biology, Helmholtz Zentrum München, German Research Center for Environmental Health, Neuherberg, Germany. [3]Technical University of Munich, Department of Mathematics, Munich, Germany. [4]Data & Analytics (D&A), Roche Pharma Research and Early Development (pRED), Roche Innovation Center Munich, Munich, Germany. [5]Large Molecule Research (LMR), Roche Pharma Research and Early Development (pRED), Roche Innovation Center Munich, Munich, Germany. [6]Roche Innovation Center Zurich, Roche Pharma Research and Early Development (pRED), Zurich, Switzerland. [7]Research and Early Development (RED), Roche Diagnostics Solutions, Roche Innovation Center Munich, Munich, Germany. [8]These authors contributed equally: Sayedali Shetab Boushehri, Katharina Essig. ✉e-mail: elke.glasmacher@roche.com; carsten.marr@helmholtz-munich.de; fabian.schmich@roche.com

formation to treat cancer and autoimmune diseases[12–15]. Although significant progress in developing immunological synapse targeting agents has been achieved in the last years[9], there is still a need to refine the compounds further, especially to improve their efficacy. It has been identified that antibody size and format[16,17], the dose, as well as target expression[18], can be critical parameters for immunological synapse formation and its effect on T cell function.

However, so far, no study has provided a tool to systematically quantify and characterize the morphology of the immunological synapse, investigate its correlation to T cell response, or identify properties predictive of the efficacy of antibodies in vitro. As a consequence, only a literature-guided set of fluorescent stainings relevant for investigating the immunological synapse is set in an otherwise untargeted approach, allowing the exploration of a broad range of possible characteristics. The key technology for high-throughput data acquisition for this purpose is imaging flow cytometry (IFC), combining the benefits of traditional flow cytometry with deep, multichannel imaging on the single-cell level[19]. IFC has recently been successfully applied to visualize and quantify the immunological synapse of primary human T:APC cell conjugates[20–22]. However, none of these studies investigated the formation of the immunological synapse in the context of T cell function.

Recent studies have demonstrated the potential of machine learning algorithms for a more robust and accurate analysis of high-throughput imaging data, an approach that has been demonstrated to overcome the limitations of conventional gating strategies[23–25]. Leveraging machine learning for IFC data analysis has also enabled the identification of morphological patterns in the cell, combining RNA and protein data analysis, and implementing predictive models[23–27]. While limited open-source software implementations designed for IFC data analysis are available[27,28], they either rely on additional third-party software which adds complexity to the analysis pipeline or focus on prediction performance only and lack explainability. synapse in the context of T cell effector function (cytokine production).

Here, we present scifAI, a machine learning framework for the efficient and explainable analysis of high-throughput imaging data based on a modular open-source implementation. We also publish the largest publicly available multichannel IFC dataset with over 2.8 million images of primary human T-B cell conjugates from multiple donors and demonstrate how scifAI can be used to detect patterns and build predictive models. We showcase the potential of our framework for (1) the prediction of immunologically relevant cell class frequencies, (2) the systematic morphological profiling of the immunological synapse, (3) the investigation of inter-donor and inter and intra-experiment variability, as well as (4) the characterization of the mode of action of therapeutic antibodies and (5) the prediction of their functionality in vitro. Combining high-throughput imaging of the immunological synapse using IFC with rigorous data preprocessing and machine learning enables researchers in pharma to screen for novel antibody candidates and improve evaluation of lead molecules in terms of functionality, mode-of-action insights and antibody characteristics such as affinity, avidity, and format.

## Results
### Comprehensive multichannel imaging flow cytometry dataset of the immunological synapse
Formation of T cell immune synapses occurs at variable and relatively low frequencies depending on the donor, the APC, and pharmacological perturbations[9,12,20–22]. Therefore, high-throughput IFC was selected as the method of choice to capture a large number of samples, enabling the detection of subtle changes in cell morphology. Using IFC, we generated a comprehensive dataset for the systematic analysis of the immunological synapse of T-B conjugates (Fig. 1a, Supplementary Figs. 1a and 2). Human memory CD4+ T cells, isolated from peripheral blood of different donors were co-cultured with superantigen

(*Staphylococcus aureus* enterotoxin A, SEA)-pulsed EBV (Epstein-Barr virus)-transformed lymphoblastoid B cells (B-LCL) expressing high levels of the co-stimulatory molecules CD86 and CD80 or left untreated (Supplementary Figs. 1b, c and 3a, b). P-CD3ζ (Y142) as a readout of early T cell activation, the highest titrated concentration of SEA (100 ng/mL), and a time point of 45 min was chosen to investigate functional immune synapses (Supplementary Fig. 1d, e). In total, we screened nine donors in four independent experiments (Supplementary Figs. 2 and 3a) and acquired 1,182,782 images (±SEA, Supplementary Fig. 3b). The designed multichannel panel consisted of brightfield (BF), F-actin (cytoskeleton), MHCII, CD3, and P-CD3ζ (TCR signaling) allowed to capture a wide range of biologically motivated, potentially relevant characteristics of the immunological synapse (Fig. 1a)[6,20]. Dead, deformed, unfocused, or cropped cells were removed using a multi-step pipeline ("Methods"). Additionally, a set of 5221 images from seven randomly selected donors was labeled by an expert immunologist (K.E.) into nine classes organized in two levels. (Fig. 1b and Supplementary Fig. 3e). The first level represented the number of existing cells in the image: singlets ($n = 1$), doublets ($n = 2$), and multiplets ($n > 2$). The second level characterizes the type of cells, their interactions with each other, and the presence of TCR signaling. The singlets are composed of "single B-LCL," "single T cell w/o signaling," and "single T cell w/ signaling" classes. The doublets include the "T cell w/ small B-LCL," "B-LCL and T cell in one layer", "synapse w/o signaling," "synapse w/ signaling," and "no cell-cell interaction" classes. The class "multi-synapse" contains more than two cells and at least one B-LCL and T cell. Even though the 'T cell w/ small B-LCL' and 'no cell-cell interaction' classes were artifacts of the experiments, they were annotated to enhance the predictive power of classification models and subsequently filtered out and not used in further analyses ("Methods").

Two donors were randomly selected to assess intra- and inter-rater variability within two experiments (donor 1 in experiment III and donor 7 in experiment IV). Next, 100 annotated samples from donor 1 and 224 annotated samples from donor 7 were randomly selected for reannotation. Four annotators with diverse backgrounds annotated the images (two immunology experts, including the original annotator and a new expert, one data analyst, and one IFC analyst). For donor 1, Cohen's kappa[29] scores were 0.84 (intra-), 0.80 (inter-), 0.79 (inter-), and 0.66 (inter-rater), respectively, compared to the original annotation by rater 1. For donor 7, comparing the original annotation with the new annotations yielded Cohen's kappa scores of 0.95 (intra-), 0.86 (inter-), 0.78 (inter-), and 0.75 (inter-rater), respectively. The strong agreement between the annotators, especially between the immunologists (scores > 0.8 are considered as almost perfect agreement[29]), gave us confidence in the reproducibility and validity of the original annotation (Supplementary Fig. 3f).

### scifAI: an explainable AI python framework for the analysis of multichannel imaging flow cytometry data
High-throughput imaging flow cytometry enables systematic profiling of millions of cells, thus providing a valuable resource for gaining biological insights[19]. Full manual annotation of such large datasets is prohibitive as expert time is scarce and expensive[30,31]. Hand-crafted gating strategies, commonly used in IFC applications[22], are hard to reproduce, often subjective and biased, and time-consuming for extensive experiments[32,33]. Additionally, it has been shown that they can be suboptimal in prediction performance[25,34]. In order to overcome these limitations, we developed the single-cell imaging flow cytometry AI (scifAI) framework for the unbiased analysis of high-dimensional high-throughput IFC data.

This open-source framework was developed in Python, leveraging functionality from state-of-the-art modules, such as scikit-learn, SciPy, NumPy, pandas, and PyTorch ("Methods"), allowing for smooth integration and extension of existing analysis pipelines. Universally

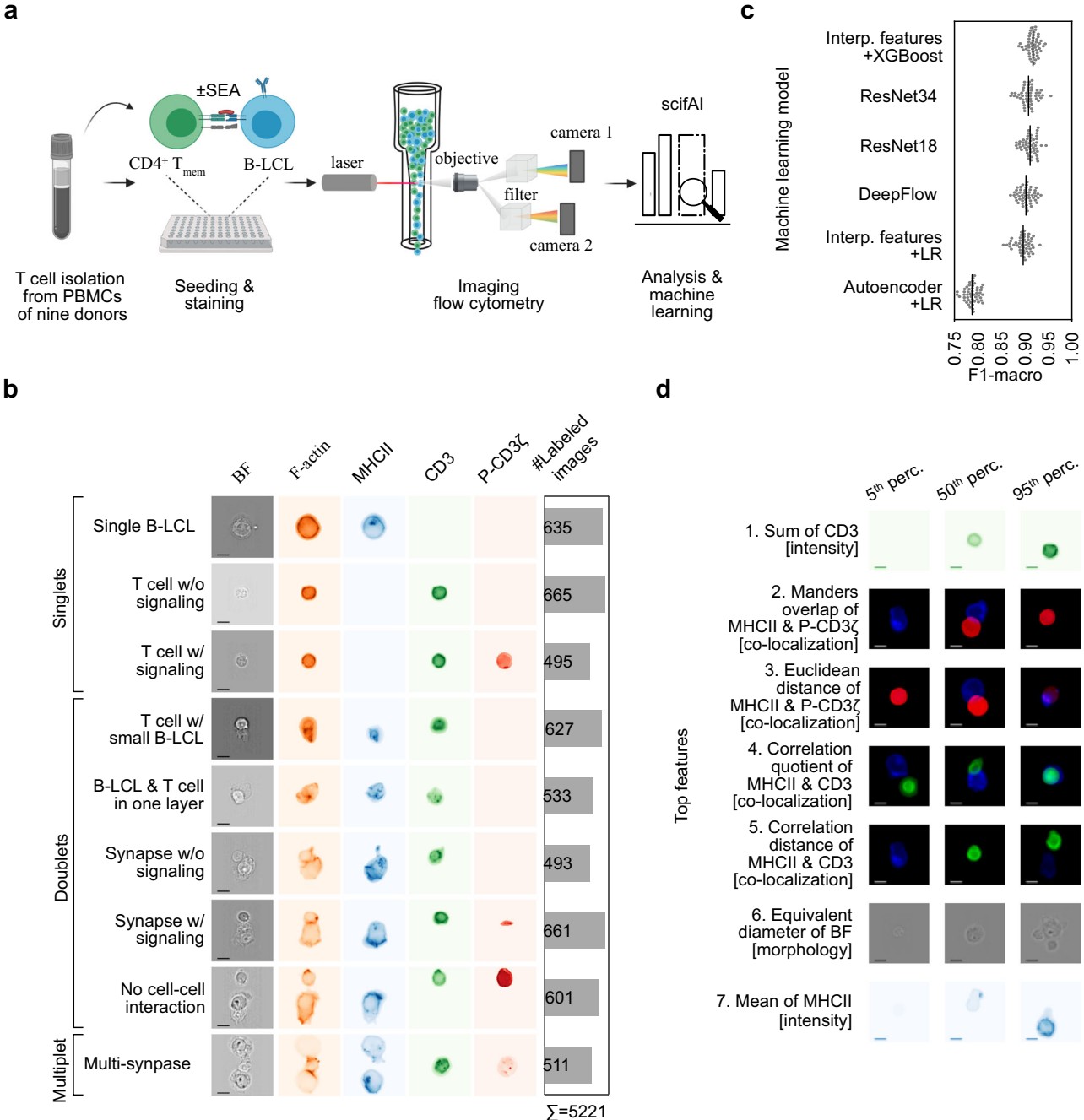

**Fig. 1 | Explainable machine learning accurately predicts immunologically relevant cell classes from IFC data and identifies the most informative image features. a** Schematic representation of the data generation and analysis pipeline. To systematically analyze the immunological synapse of T-B conjugates, 1,182,782 images were acquired with an imaging flow cytometer. Next, scifAI was used to extract morphological features, train machine learning models, profile immunological synapses, and characterize the functionality of therapeutic antibodies. **b** A subset of 5221 images was annotated by an expert into nine immunologically relevant classes that can be grouped into singlets (either B or T cells), doublets (with one B and one T cell), and multiplets (containing >2 cells). Cell images show brightfield (BF, scale bar = 2.4µm), F-actin (cytoskeleton), MHCII, CD3, and P-CD3ζ (a marker for TCR signaling). **c** Six different approaches to training predictive machine learning models for identifying the immunologically relevant classes were

benchmarked, combining different classification algorithms and feature engineering strategies. These approaches included interpretable (interp.) features combined with explainable classifiers, an autoencoder to generate data-driven features, an explainable classifier, and three convolutional neural networks. Interpretable features combined with the XGBoost classifier resulted in the best trade-off between interpretability and classification performance. Each point depicts an iteration in a stratified 5-fold cross-validation scheme with 10 times repetition. The black line represents the mean of the points. Source data are provided as a Source Data file. **d** Top seven features for detecting cell classes were ranked based on in-model feature importance ("Methods"). The features include co-localization and intensity of MHCII (blue), CD3 (green), and P-CD3ζ (red), as well as the morphology of BF (gray). The exemplary images are sampled from the 5th, 50th, and 95th percentile of the distribution of each feature (scale bar = 2.4µm).

applicable for single-cell brightfield or fluorescent imaging projects, the framework provides functionality for the import and preprocessing of input data, several feature engineering pipelines, including the implementation of a set of biologically motivated features and

autoencoder-generated features ("Methods"), as well as the methodology for efficient and meaningful feature selection. Moreover, the framework implements several machine learning and deep learning models for training supervised image classification models, e.g., for

predicting cell configurations such as the immunological synapse. The implemented models included logistic regression[35], linear discriminant analysis[35], support vector machine[35], random forest[36], and XGBoost[37], as well as deep learning models including a multi-encoder autocoder[25], DeepFlow[24], ResNet18 and ResNet34[38]. Following the principle of multiple-instance learning, the framework also implements functionality to regress a set of selected images against a downstream continuous readout, such as cytokine production. Extensive documentation, as well as how to reproduce the analysis in the form of Jupyter notebooks, is provided online at https://github.com/marrlab/scifAI/ and https://github.com/marrlab/scifAI-notebooks.

scifAI's explainable design follows the definition of Singh et al.[39] and Tjoa et al.[40], where machine learning models are explained by either in-model mechanisms, such as Gini-index or gain, or by using post-model methods, such as saliency maps[41] for deep learning models. Using interpretable features with a model whose internal mechanisms can be readily analyzed, scifAI natively provides in-model explainability and is fully compatible with other libraries, such as SHAP[42] or Captum[43], to provide further post-model explainability.

## scifAI enables high-throughput profiling of the immunological synapse

In order to characterize the immunological synapse in an unbiased fashion, we first designed and computed a series of biologically motivated, interpretable features using the scifAI framework. These features were based on morphology, intensity, co-localization, texture, and synaptic features extracted from the 5-panel stained images and their corresponding masks ("Methods" and Supplementary Fig. 4a–c). Morphology features included shape-related characteristics such as area, perimeter, and roundness of cells. Intensity and texture features were implemented to quantify the existence, distribution, and regularity of intensities within cells[44]. Co-localization features were designed to capture similarities and differences in intensities among different channels[45]. Finally, synaptic features were implemented to give insights into the distribution of intensities within the synapse region compared to the rest of the cell[33,46]. Synaptic features were implemented based on the signal intensity ratio of each fluorescent channel in the synaptic area to the whole cell. These features allowed for comparing cell states within and among different populations. Leveraging the large amount of unlabeled data, we also implemented a multichannel convolutional autoencoder to learn a second set of data-driven features from the images in an unsupervised fashion[25]. The autoencoder was designed to encode the images to a 256-dimensional abstract feature space by reconstructing the input images. Considering the computational complexity of machine learning modeling on high-throughput data, all mentioned algorithms were implemented using parallel computing, fully leveraging high-performance computing (HPC) infrastructure to speed up the calculations by distributing the computational load on multiple CPUs ("Methods").

Subsequently, scifAI was used to compose a supervised machine learning pipeline for the classification of the 5221 annotated images across the nine immunologically relevant cell classes. We trained and benchmarked a series of supervised machine learning models for the prediction of all nine classes using both the interpretable feature space as well as the abstract autoencoder features across all donors and experimental conditions. The models included an XGBoost[37] classifier on the interpretable features and a multi-class logistic regression (LR) on the interpretable and data-driven features. To pre-select the features and reduce the dimensionality, we implemented a feature pre-selection pipeline using an ensemble of different methods ("Methods" and Supplementary Fig. 5a, b). We also trained a number of convolutional neural network (CNN) architectures, such as Resnet18, ResNet34, and DeepFlow, which had previously been shown to be successful in classification tasks on imaging flow cytometry data[23–25]. For Deepflow, a random initialization was used. A self-supervised method called Barlow Twins[47] was used to pre-train ResNet18 and ResNet34. Here, the unlabeled part of the ±SEA-based was utilized based on Barlow Twins self-supervision task, and then the weights were transferred for the supervised training. This method has been shown to improve the performance of the CNN models in classification tasks[47]. In the supervised training, the CNN architectures intrinsically learned a feature representation based on the input images and their corresponding labels. To estimate the performance of the models, the annotated ±SEA-based dataset with 5221 images in total was split into train (70%) and test (30%) sets, resulting in 3654 images for training and 1567 images for testing. All models were trained on the stratified ±SEA-based training set. We compared the macro F1-score on the ±SEA-based hold-out test set to benchmark the classification model and feature space combinations as the ("Methods"). The XGBoost model using the interpretable feature set performed best (F1-macro = $0.92 \pm 0.01$, mean ± std 5-fold cross-validation with 10 repetitions) among all the classifiers. It was followed by convolutional neural networks ResNet34 ($0.91 \pm 0.01$), ResNet18 ($0.91 \pm 0.02$), and DeepFlow ($0.90 \pm 0.02$). They were followed by multi-class logistic regression using the interpretable feature set ($0.90 \pm 0.02$) and logistic regression using the data-driven feature set ($0.79 \pm 0.02$). Based on the performance and explainability, the XGBoost model was selected as the final classifier for label expansion to the full dataset (Fig. 1c). Investigation of the model's confusion matrix on the hold-out set revealed that misclassifications occurred mostly within the cell classes' signaling property, whereas all other classes showed good overall concordance (Supplementary Fig. 5c).

An interesting observation was that even though the autoencoder has been trained on all unlabeled data and thus was allowed to learn a data-driven representation of the full dataset, this derived feature space is considerably less performant as compared to the engineered interpretable features in a logistic regression model ($0.79 \pm 0.02$ vs. $0.90 \pm 0.02$, Fig. 1c). To confirm the quality of the interpretable features, the previously trained XGBoost and logistic regression models were compared with different classifiers in an ablation study on the interpretable features. These classifiers included random forest, support vector machine, and linear discriminant analysis. While XGBoost performed best ($0.92 \pm 0.02$), we only observed minor drops in performance using random forest (RF, $0.90 \pm 0.02$), linear regression (LR, $0.90 \pm 0.02$), support vector machine (SVM, $0.90 \pm 0.02$), or linear discriminant analysis (LDA, $0.87 \pm 0.02$). Thus, we concluded that the predictive performance driving factor is the feature space of interpretable features (Supplementary Fig. 6a).

After confirming the quality of the interpretable features and the choice of the classifier, it was investigated whether they can be used in real-world examples where it is desirable to have a model that can be generalized to new donors. Therefore, leave-one-donor-out cross-validation was performed using the interpretable features and XGBoost. The cross-validation yielded F1-macro values of $0.88 \pm 0.04$, demonstrating good generalizability across donors (Supplementary Fig. 6b).

Next, we focused on the explainability of the pipeline and explored which underlying features drive the class prediction, ranking features by their respective feature importance. The feature importance was based on the in-model mechanism called gain[37], which signifies the relative contribution of a corresponding feature to the classification ("Methods"). The most predictive features were based on the intensity of CD3, co-localization of MHCII & P-CD3ζ, co-localization of MHCII & CD3, the cell morphology in BF, and the intensity of the MHCII (Fig. 1d). The intensities of CD3 and MHCII imply the presence of a T cell or B-LCL in the image. The co-localization of MHCII and P-CD3ζ measures how many overlapping pixels the two proteins share. For example, 'synapse w/ signaling' has a lower overlap between the T cell and the B-LCL than the 'T cell & B-LCL in one layer,' and thus a lower co-localization of MHCII and P-CD3ζ. The same logic can be applied to the

co-localization of MHCII and CD3. The equivalent diameter of BF hints at the size of the cells or the existence of multiple cells. Based on the features and the definition of classes, one could speculate that the classifier uses (1) the intensity of CD3 and MHCII to detect the existence of T and B-LCL cells in the image, (2) the co-localization of MHCII & P-CD3ζ as well as MHCII & CD3 the to detect the different doublets types and the existence of signaling (3) the cell morphology in BF to assist detecting the cell type (Fig. 1d). To validate the feature importance with a widely used, post-model explainability approach, SHAP[42] values were calculated ("Methods"). SHAP is a game theory-based approach that has recently been shown to be the most used post-model explainability method in related topics[48]. The top-5 SHAP values (Supplementary Fig. 6c) were similar to the previously shown feature importances based on model-intrinsic gain, providing additional confidence in the stability of the feature importance estimation (Fig. 1d).

Finally, to confirm the quality of experimental data, it was investigated whether adding SEA led to higher synapse frequency. The trained XGBoost model was applied to the ±SEA-based complete dataset. After data cleaning and removing artifacts ("Methods"), the frequency of synapses was calculated for the -SEA and +SEA cases. The results showed that adding SEA significantly increased 'synapses w/ signaling' frequency from $0.95\% \pm 0.55\%$ to $2.58\% \pm 0.86\%$ ($p < 0.001$, Mann-Whitney U test, $n = 9$ donors). Therefore, the experimental data followed the expected biological behavior. Moreover, the low percentage of the 'synapses w/ signaling' confirmed the rarity of synapses and the importance of a high-throughput technology while working on synapse formation.

## A subset of annotated data and available IFC channels suffices for a high classification performance

Considering that manual annotation of images can be time-consuming, we performed an ablation study to investigate how many annotated samples were necessary to achieve a high classification performance. We repeatedly trained the model on stratified subsets of the ±SEA-based training data (5%, 15%, ..., 95%) and evaluated the F1-macro on the ±SEA-based test set. The results showed that by using 1500 images (45% of the training data), we could achieve $0.90 \pm 0.01$ F1-macro on the test set (Supplementary Fig. 7a). This result demonstrated that it is possible to halve the manual annotation time and still achieve a similar quality of classification performance, as compared to using the whole annotated set ($0.92 \pm 0.01$).

Next, we investigated the effects of fluorescent channels on the classification performance to determine which antibodies are essential for the detection of synapses and which ones can be freely exchanged depending on the biological context. Considering that BF is a stain-free channel provided for free by IFC, we kept the BF channel fixed and added all possible combinations of the fluorescent channels to train the model. We found that the combination of the channels BF, MHCII, and P-CD3ζ sufficed to reach an F1-macro of $0.91 \pm 0.01$ (Supplementary Fig. 7b), similar to using all the channels ($0.92 \pm 0.01$).

## Characterizing the impact of therapeutic antibodies on synapse formation

We next used scifAI to investigate the effects of therapeutic antibodies on the formation of the immunological synapse and to characterize their morphological profiles better. This analysis included the investigation of potential class frequency changes and feature differences. We chose two antibodies, one activator and one inhibitor of immune responses. The activating T cell bispecific (TCB) antibody was designed to target CD3 and CD19, a co-receptor of B cells[49] (Fig. 2a). The inhibitory antibody, Teplizumab, is described to only bind to CD3 (Fig. 2c) and has been shown to dampen T cell responses[50,51]. For each antibody, an appropriate control (Ctrl-TCB and isotype) was run within the same experiment and donor. Since Teplizumab required an existing immune response for subsequent

inhibition, we used SEA to first stimulate the T cells (Fig. 2c). The same setup was also used for the isotype control. Six donors across two experiments for CD19-TCB and seven donors across three experiments for Teplizumab were measured (Fig. 2b, d and Supplementary Fig. 3c, d) and were used to predict the class for all images based on the interpretable features. A data cleaning pipeline was also implemented to filter out unwanted images such as experimental artifacts ("Methods" and Supplementary Fig. 8). To ensure that the previously trained XGBoost model was transferable from ±SEA to the antibody experiments, an expert (K.E.) annotated a randomly selected subset of 396 images for CD19-TCB and 227 images for Teplizumab. A high concordance between ground truth annotations and XGBoost predictions on the new experiments (macro F1-score = 0.86 for TCB and 0.85 for the Teplizumab) confirmed that the trained model generalizes across experiments and can thus be utilized for further analyses (Supplementary Fig. 9). For a compact representation of class frequency changes, we computed log2-fold changes between the antibodies and their controls. In a second step, we focused on the feature differences of synapses under antibody stimulation and selected all the images predicted as 'synapses w/ signaling' for each donor and compared interpretable features from only fluorescent channels, including texture, synaptic features, morphology, intensity and co-localization between antibodies and their controls ("Methods"). Considering that we were interested in the mode of action of antibodies, we focused on fluorescent channels providing targeted information on components of the cell expected to change during synapse formation morphologically.

## CD19-TCB increases the formation of stable immune synapses

Stimulation of the immune response by CD19-TCB led to a significant increase of doublets and multiplets frequencies. The 'synapse w/ signaling' class showed thereby the highest increase (median $\log_2$(CD19-TCB/Ctrl-TCB) = 2.7, $n = 6$ donors, $p = 0.036$) followed by 'multisynapse' (median = 2.03, $p = 0.036$), 'B-LCL & T cell in one layer' (median = 1.99, $p = 0.036$), and 'synapse w/o signaling' class (median = 0.59, $p = 0.036$). For the singlets, the overall trend was a decrease in the class frequency of 'single B-LCL' (median = −0.21, $p = 0.036$) and 'single T cell w/o signaling' (median = −0.77, $p = 0.036$) (Fig. 2e).

Next, we investigated the feature differences in synapses induced by the CD19-TCB ("Methods"), comparing the 210 interpretable features from all fluorescent channels. We found 210 significantly increased and 163 significantly decreased features out of 210*6 = 1260 possibilities from combination of features and donors (Fig. 3a). All donors exhibited mostly similar responses toward the stimulation with CD19-TCB. On average, $42 \pm 4$ features were significantly decreased, and $39 \pm 11$ features were significantly increased per donor (dashed lines bottom Fig. 3a). From these features, we were able to find a number of features with similar changes within at least 4 out of 6 donors (Fig. 3a and Supplementary Table 1). We also observed an increase in 'mean intensity of P-CD3ζ' with higher enrichment within the synaptic area (Fig. 3b, c and Supplementary Table 1). In addition, we also detected a stronger enrichment of F-actin and MHCII toward the synapse (Fig. 3d–g). Taken together, the observed increase in doublet and multiplet frequencies as well as a stronger enrichment of F-actin and MHCII in the synaptic area, indicated an enhanced formation of tight immunological synapses, translating into an efficient TCR signaling. These observations align with the mode of action already described in general for TCBs, promoting a stable interaction between tumor cells and T cells[15,52,53].

## Teplizumab alters synapse formation and TCR signaling

In contrast to the CD19-TCB, treatment with Teplizumab reduced the frequency of doublets and multiplets significantly (Fig. 2f). The highest decrease was observed for the 'synapse w/ signaling' class (median $\log_2$(Teplizumab/Isotype) = −0.75, $n = 7$, $p = 0.022$), followed by

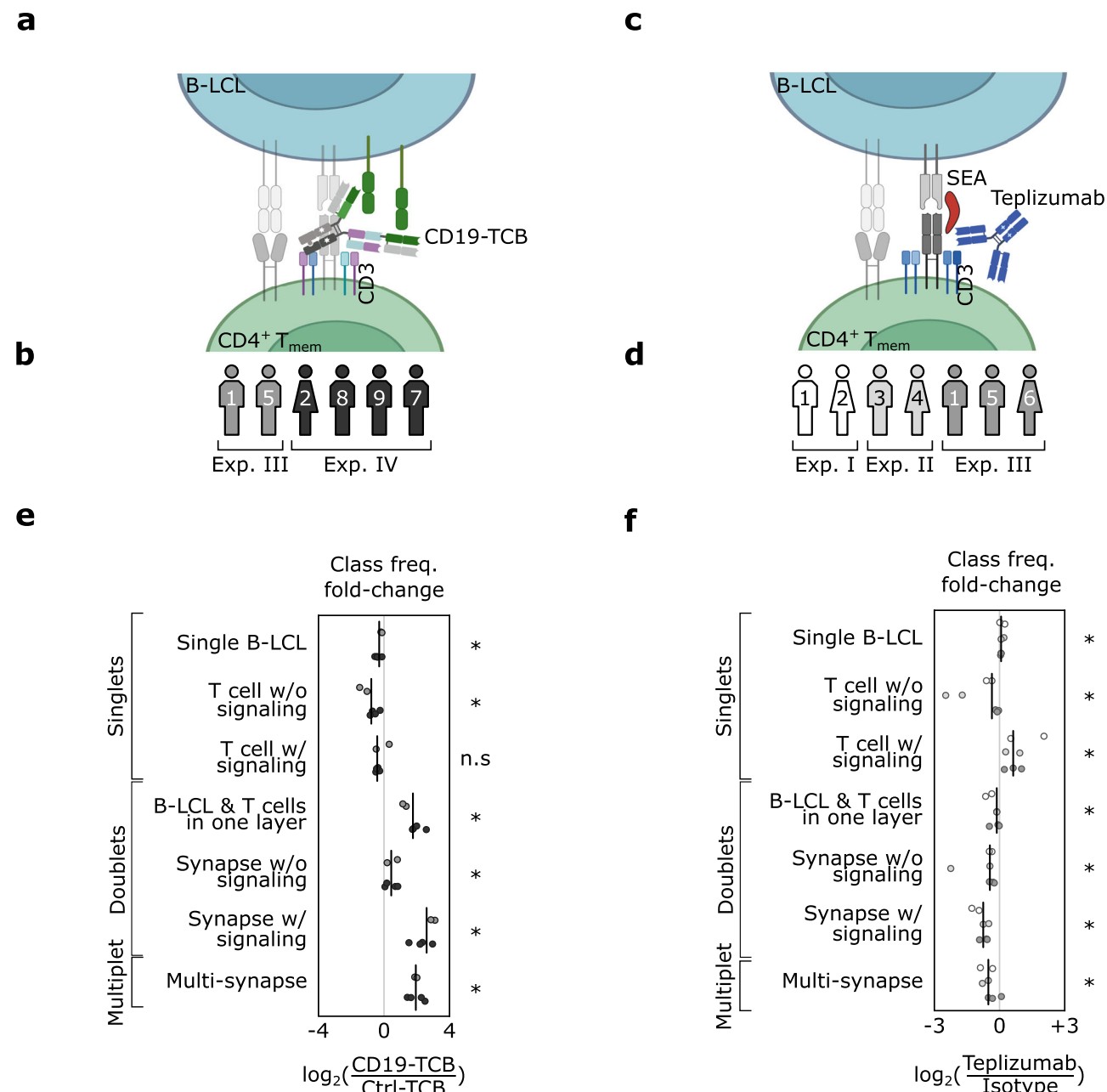

**Fig. 2 | CD19-TCB and Teplizumab show significant changes in the frequencies of synapses. a, c** Schematic representation of the mode of action of CD19-TCB and Teplizumab. CD19-TCB binds with one arm to the T cell receptor CD3 and with two arms to the B cell co-receptor CD19, thereby bringing T and B cells into proximity and activating T cells. Teplizumab binds with two arms to CD3 and has been described to inhibit T cell activation. The T cells needed to be first stimulated via the superantigen SEA to exert their full suppressive function. **b, d** Donors and their respective experiments were used for the class frequency analysis in Fig. 2. **e, f** Feature difference analysis in Fig. 3, and cytokine prediction analysis in Fig. 4. **e, f** Class frequency differences depicted as log2-fold changes between CD19-TCB (6 donors) or Teplizumab (7 donors) and their corresponding controls (Ctrl-TCB & isotype). Each dot represents a donor color-coded as in (**b**) or (**d**). The vertical black line is the median across donors for each class. A two-sided Wilcoxon-rank-sum was used to analyze the significance of the log2-fold-change, and the $p$-values were corrected using the Benjamini-Hochberg procedure. For (**e**), the $p$-values are 0.036, except for 'T cell w/ signaling', which was 0.109. For (**f**), the $p$-values are 0.022, 0.022, 0.022, 0.109, 0.022, 0.022, 0.036. (*) represents $p$-value < 0.05, and (n.s.) represents not significant. Source data are provided as a Source Data file.

'multiplets' (median = −0.69, $p = 0.036$), and 'synapse w/o signaling' (median = −0.57, $p = 0.022$). Accordingly, 'single T cell w/ signaling' (median = 0.70, $p = 0.022$) and 'single B-LCL' (median=0.09, $p = 0.022$) were increased significantly as compared to the isotype. Surprisingly, the 'T cell w/o signaling' class frequency was significantly decreased (median = −0.30, $p = 0.022$), probably due to the significant increase of 'single T cell w/ signaling' (Fig. 2f).

We next investigated feature differences in synapses induced by Teplizumab in seven donors ("Methods"). From 'synapses w/ signaling'

images we extracted 132 features based on F-actin, MHCII and P-CD3ζ and their co-localizations. CD3 features could not be included in the analysis because the binding of Teplizumab and the anti-CD3 staining antibody interfere, therefore an anti-CD4 staining antibody was used to identify T cells. We found 131 significantly increased and 169 significantly decreased features out of 132*7 = 924 possibilities (Fig. 3h and Supplementary Table 2). In particular, on average, Teplizumab led to 20 ± 15 significantly decreased features and 23 ± 9 significantly increased features per donor (dashed lines bottom Fig. 3h).

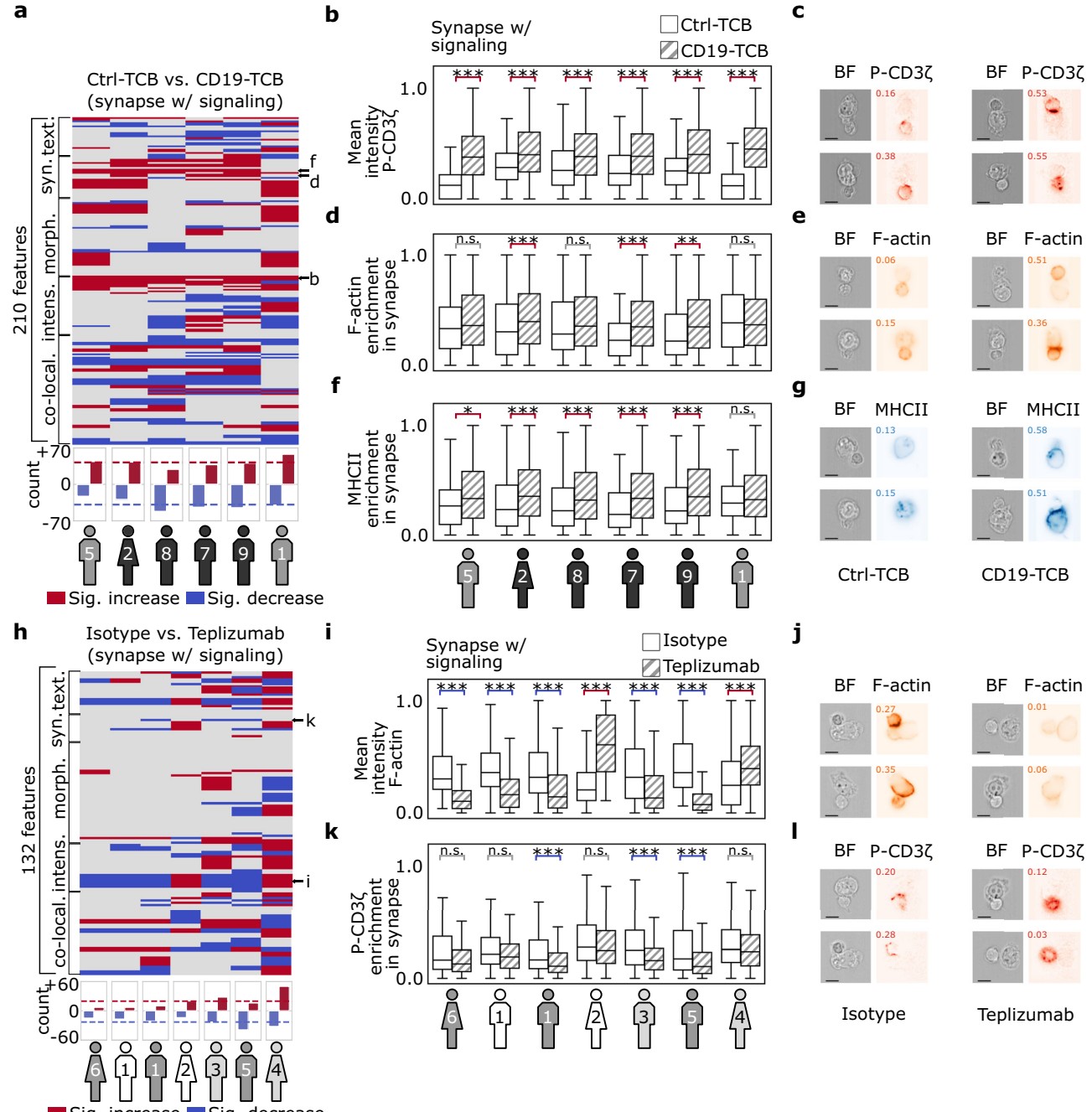

Donor 6 showed the least number of changes with 15 significantly increased features. In contrast, donor 4 yielded the highest number of increased features with 32 features, indicating a fair amount of inter-donor variability. We found a set of features that were significantly increased or decreased for at least 3 out of 7 donors (Fig. 3i, k). We observed a decrease in the mean intensity of F-actin, whereas donors 2 and 4 indicated a significant increase (Fig. 3i, j). This opposite reaction of the two donors could also be detected for other F-actin-related features (Supplementary Table 2). Besides the changes in F-actin features, we also detected a significant reduction of P-CD3ζ intensity within the synapse and observed a stronger clustering of TCR signaling around the whole T cell (Fig. 3k, l). In conclusion, we gained new insights into the immunosuppressive mode of action of Teplizumab as we observed a reduction in the number of synapses as well as changes in the F-actin reorganization and P-CD3ζ signaling toward the synapse.

## Morphological profiles of the immunological synapse predict functionality of therapeutic antibodies in vitro

Next, the capabilities of scifAI in predicting the functionality of antibodies by analyzing T cell cytokine production were explored. We included an additional antibody in our analysis, the CD20-TCB (Supplementary Fig. 3c and "Methods"). CD20-TCB is a therapeutic antibody with the same format similar to CD19-TCB but with varying target moiety[15]. CD3 features could not be included in this analysis because the binding of the TCBs interferes with the anti-CD3 staining antibody resulting in erroneously lower CD3 intensity features. Given the richness of interpretable features, we investigated whether it is possible to forecast downstream T cell responses as measured by Granzyme B (GrzmB) for the TCBs. While the IFC measurement was taken after 45 min, GrzmB was measured after 24 h, respectively, using conventional fluorescence-activated cell sorting (FACS) for each donor and condition to address the effects of the antibodies in later time points

**Fig. 3 | CD19-TCB and Teplizumab induce morphological changes in synapse formation, including texture, intensity, and synaptic features. a** Systematic comparison of 210 relevant features between CD19-TCB and Ctrl-TCB across images predicted as 'synapse w/ signaling' across six donors. Each line represents a feature, and each column represents a donor. For each donor, the significantly increased features are depicted in red, and the significantly decreased ones are in blue. The significance is measured by a two-sided Mann-Whitney U test and corrected by the Benjamini-Hochberg procedure. In each column, the number of 'synapse w/ signaling' for every donor is: donor 5 experiment 3 (Ctrl-TCB = 86, CD19-TCB = 660), donor 2 experiment 4 (265, 729), donor 8 experiment 4 (194, 753), donor 7 experiment 4 (96, 822), donor 9 experiment 4 (169, 746) and donor 1 experiment 3 (77, 666). The donors are sorted based on the number of significantly changed features. The bottom barplot shows the count of increased or decreased features per donor. Heatmap rows with arrows are shown in detail in (**b**, **d**, and **f**). **b**, **d**, **f** Statistical and visual inter-donor comparison of three representative features between CD19-TCB and Ctrl-TCB. The features are mapped separately between zero and one for each donor for visualization purposes. The boxes represent quartiles of the data, and the whiskers represent a 1.5 times interquartile range from the nearest hinge. The $p$-values for (**b**) are 7.7e-20, 8.7e-11, 7.2e-08, 3.3e-08, 4.8e-12, and 5.7e-24. For (**d**), they are 3.8e-01, 2.4e-04, 1.4e-01, 3.9e-05, 9.1e-04, and 6.5e-01. For (**f**) we have 3.8e-02, 1.5e-07, 1.9e-04, 3.8e-05, 1.9e-05, and 2.7e-01. (*) represents $0.01 < p$-value $< 0.05$, (**) represents $0.01 < p$-value $< 0.01$ and (***) represents $p$-value $< 0.001$. **c**, **e**, **g** Visual representatives for all features were randomly sampled for both Ctrl-TCB and CD19-TCB from donor 9 and were found to be in concordance with the statistical results (scale bar = 2.4μm). **h** Systematic comparison of 132 relevant features between Teplizumab and isotype across images predicted as 'synapse w/ signaling' among all six donors. The significance is measured by a two-sided Mann-Whitney U test and corrected by the Benjamini-Hochberg procedure. The color code, barplot, and sorting are the same as described in (**a**). In each column, the number of 'synapse w/ signaling' for every donor is: donor 6 experiment 3 (isotype = 204, Teplizumab = 128), donor 1 experiment 1 (254, 119), donor 1 experiment 3 (328, 222), donor 2 experiment 1 (265, 89), donor 3 experiment 2 (421, 288), donor 5 experiment 3 (326, 227), donor 4 experiment 2 (480, 285). **i**, **k** Statistical and visual inter-donor comparison of two representative features between Teplizumab and its isotype (also shown by three small arrows in **h**). The $p$-values for **i** are 4.3e-24, 8.7e-15, 4.4e-15, 4.3e-22, 3.0e-12, 1.4e-52, 5.2e-13. The boxes represent quartiles of the data, and the whiskers represt 1.5*interquartile range from the nearest hinge. The $p$-values for (**k**) are 9.0e-02, 1.4e-01, 7.5e-06, 4.1e-01, 1.8e-08, 1.8e-05, and 1.5e-01. (*) represents $0.01 < p$-value $< 0.05$, (**) represents $0.01 < p$-value $< 0.01$ and (***) represents $p$-value $< 0.001$. **j**, **l** Visual representatives for two features were randomly sampled for both isotype and Teplizumab from donor 3 and were found to be in concordance with the statistical results (scale bar = 2.4μm). Source data are provided as a Source Data file.

(Supplementary Fig. 10a). In line with the differences in target expression, the CD20-TCB ($20.23 \pm 6.32$, $n = 4$) showed the highest expression of GrzmB, followed by the CD19-TCB ($12.58 \pm 3.04$) and Ctrl-TCB ($1.16 \pm 0.51$) (Fig. 4a and Supplementary Fig. 10b). A similar pattern was also detected for killing of two tumor cell lines with different expression levels of CD19 and CD20 (Supplementary Fig. 10c, d).

Since there is a one-to-many relationship between FACS cytokine measurements and IFC images, where each cytokine measurement corresponds to an IFC cell population consisting of 54,708 images on average, an aggregation pipeline based on the interpretable features was implemented. For each donor and condition, the first images predicted as synapses were selected, and their previously extracted features were aggregated using the 5th, 50th, and 95th percentile. This aggregation ensured that every feature's extrema and average expression were captured (Fig. 4b and "Methods"). Next, it was attempted to predict the cytokines for an unseen antibody. Due to the low number of samples, a linear model with Lasso Lars penalization was used ("Methods"). The cross-validation was performed on CD19-TCB and CD20-TCB, while the No Ab and Ctrl-TCB were kept in the training set (Fig. 4c). The prediction performance reached a Spearman correlation of 0.38 (Fig. 4d). While we could observe subtle differences between the predictions and the ground truth, the model correctly identified the separation between CD20-TCB and CD19-TCB. Furthermore, the model suggested that the 'standard deviation of MHCII (95th perc.) [intensity]' and 'eccentricity of F-actin (95th perc.) [morphology]' as the most important features (Fig. 4e, f).

## Discussion

In the present work, we established scifAI, a pipeline based on two innovative technologies, imaging flow cytometry and explainable machine learning to understand the mode of action and predict the functionality of therapeutic antibodies in vitro. We analyzed morphological profiles of the immunological synapse to better characterize the mode of action of therapeutic antibodies early after the initiation of an immune response and to apply it to forecast downstream T cell responses. This work is the first coherent functional study using large image datasets and the consequential analytical part to visualize, understand and study synapse formation and its link to predictive features.

We generated the largest publicly available imaging flow cytometry data with over 2.8 million images using human primary immune cells from nine donors in four independent experiments that were treated with various therapeutic antibodies to study synapse formation. The large number of acquired images across multiple experiments provided sufficient statistical power to enable the study synapse formation, a potentially low-probability event, going beyond previous works that did, for example, not consider inter-experiments effects or inter- and intra-donor variability[22,33]. While we demonstrate good generalizability of our model, we do not expect that the model trained on our specific IFC data will be predictive without additional training on different datasets obtained with other IFC machines out of the box due to the inherent domain shifts in resolution, magnification, light wavelengths, or focal depth. However, exploiting transfer learning, self-supervised pre-training[54], and domain adaptation and generalization techniques[55], this dataset will be a valuable resource for future applications, such as transfer to other fluorescent imaging modalities, for example, confocal microscopy, where data can be scarce. Training deep generative models to understand the mode of action of therapeutic antibodies[56] is another exciting avenue for future research.

To detect and study immunological synapses, we implemented an interpretable feature extraction and machine learning framework in Python by only using well-maintained Python modules. scifAI natively implements parallel computing, fully leveraging modern HPC infrastructure and allowing for efficient processing and analysis of high-throughput data by parallelizing tasks such as data preprocessing and model fitting across multiple CPUs. On a 24-CPU machine, scifAI enables feature extraction and class-label prediction of approximately 250,000 images per hour. This is, by orders of magnitude, more efficient than manual annotation, as reported by our annotators, with a rate of approximately 100 images per hour. These choices guarantee performance, scalability, and reproducibility and facilitate the deployment into existing workflows, which differs from previous works that use a combination of CellProfiler, R, and Python for each stage of the analysis[28,57,58]. In this work, we followed the definitions of Singh et al.[39] and Tjoa et al.[40] in the context of the explainability of machine learning and AI. We regard the biologically motivated features as interpretable as they are meaningful and can potentially hint at underlying biological mechanisms. We also considered our XGBoost model explainable as it natively provides a feature importance measure[42]. This design also enables a deeper understanding of the model's decision-making using additional methods, such as SHAP[42]. The insights from the interpretable features and explainable models have the potential to generate new biological hypotheses, which can lead to a better understanding of the underlying mechanisms at play. It

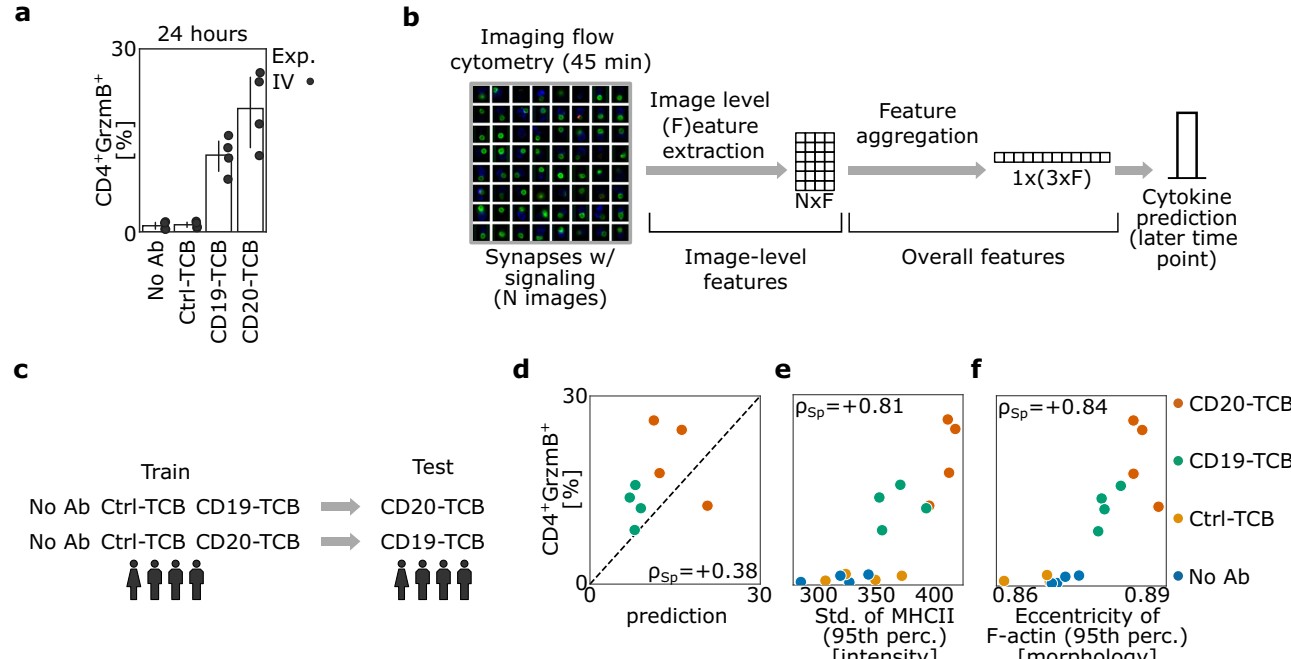

**Fig. 4 | Morphological profiles of synapses are predictive of the functionality of antibodies in vitro. a** Frequencies of GrzmB⁺ CD4⁺ T cells (gated as single live⁺ cells) measured by FACS after 24 h. Corresponding to each imaging flow cytometry experiment, a separate FACS experiment was performed with the same batch to obtain the cytokine production values. Each dot represents a donor from experiment IV (4 donors). The barplot represents the mean with a 95% confidence interval. **b** Data aggregation pipeline: For each donor and condition, images identified as synapses with and without signaling are selected ($N$ = number of detected images). Then, image-level features ($F$ = number of relevant features) are extracted across all selected images. The features were aggregated for each donor and condition using the 5th, 50th, and 95th percentile. This 620-dimensional aggregated feature reduces the cytokine prediction to a multivariate regression task. **c** Analysis scheme: Cytokines are predicted for an unseen activator antibody using only the control (No Ab and Ctrl-TCB) and another activator antibody. Data originates from one experiment with four donors. **d** Scatterplot of the predictions versus ground truth values of GrzmB⁺ CD4⁺ frequencies. The dots are only based on the predictions from (**c**); therefore, only CD19-TCB and CD20-TCB are depicted. **e, f** Scatterplot of the features 'standard deviation of MHCII (95th percentile)' and 'eccentricity of F-actin (95th percentile)' versus the GrzmB⁺ CD4⁺ frequencies. These two features were selected based on the trained linear model in (**d**). Both features show a high correlation with respect to GrzmB⁺ CD4⁺ frequencies. Source data are provided as a Source Data file.

is essential to mention that explainable models can offer a helpful intuition only if trained on interpretable features and will fail to provide a meaningful interpretation when trained on abstract features such as the bottleneck layer of an autoencoder. The combination of interpretable features and explainable machine learning enabled us to identify various relevant classes, such as immunological synapses, with state-of-the-art accuracy. It also allowed us to investigate the morphological profiles of the immunological synapse in an unbiased way and characterize the mode of action of antibodies in a biologically relevant context. This methodology is thus a substantial contribution to the field, which has so far primarily focused on performance over interpretability by using a ResNet CNN architecture as the backbone[27,54]. A potential limitation in this work is the choice of markers. While the markers were carefully selected to reflect immunological synapse characteristics, our results are restricted to those choices. Nonetheless, feature engineering in scifAI is designed in an unbiased way toward any IFC dataset with an arbitrary set of stainings. As proof of concept, we provide three examples within the scifAI code repository on how to analyze IFC datasets from Jurkat cells (3 channels per image)[24], white blood cells (12 channels)[23], and apoptotic cells (2 channels)[25] using scifAI.

To demonstrate the capabilities of the scifAI framework, we investigated the effects of two therapeutic antibodies on the immunological synapse, the CD19-TCB and Teplizumab, which are both binding CD3 and have been described to activate and suppress T cell responses, respectively[49–51]. We found that the CD19-TCB forms more stable immune synapses, as indicated by a stronger enrichment of MHCII and F-actin within the synapse, paralleled by a higher intensity of P-CD3ζ. The formation of stable T cell-tumor cell synapses has

already been reported for other TCBs like the CEA- and CD20-TCB[15,52]. In contrast to the CD19-TCB, treatment of Teplizumab yielded a decrease in synapse formation and prevented F-actin reorganization as well as localization of P-CD3ζ toward the synapse. These observations gave new insights into Teplizumab's immunosuppressive mode of action that has rarely been investigated in vitro so far[50,51]. One could speculate that steric hindrance by Teplizumab prevented T-B cell interactions leading to less stable synapses and reduced cytokine production. It has been shown that antibody size and format can substantially impact synapse formation[16,17]. Another hypothesis could be that the binding of Teplizumab induced strong TCR internalization that led to diminished SEA-mediated TCR-MHCII crosslinking and thus inhibited T cell activation. The reduced P-CD3ζ intensity in the synaptic area and the observed unpolarized distribution of the P-CD3ζ signal around the whole T cell could also indicate altered TCR signaling that might be translated into a reduced T cell effector function. High numbers of peripheral P-CD3ζ microclusters have already been reported for self-reactive T cells with altered synapse formation and aberrant T cell responses[6]. Interestingly, scifAI identified features within the synapse class, revealing inter-donor variability upon stimulation with the different antibodies. However, we were not able to correlate the variability of those features to a different functionality in vitro because the differences in the T cell responses between donors were just minor. Patient material from ongoing clinical trials could help elaborate further if these synapse features could predict clinical response. In that case, scifAI could enable us to rapidly screen for responders in vitro and potentially pre-select suitable patients for clinical trials. Taken together, by applying scifAI we were not only able to thoroughly investigate the mode of action of therapeutic antibodies

by identifying significant features but also to gain more insights into inter-donor variability that might potentially translate into different functional outcomes in vivo.

The immunological synapse has previously been studied using high-content cell imaging on human cell lines and primary cells with an artificial APC system that utilized plate-bound ICAM-1 and stimulatory antibodies[58]. Although German et al. convincingly demonstrated the capabilities of their pipeline by profiling the immunological synapse, they did not investigate whether they could use these profiles in predicting drug effectiveness[58]. In other studies, the potential of synapse formation was also investigated for CAR T cell therapy, where investigators used the mean intensity of stainings such as F-actin and P-CD3ζ per cell, clustering of tumor antigen and polarization of perforin-containing granules as a measure of synapse formation quality. These features varied between different CAR T cells and correlated with their effectiveness in vitro and in vivo as well as with clinical outcomes[59,60]. In our work, we improved this by incorporating 296 biologically motivated features such as texture, intensity statistics and synaptic-related features.

This work is the first to use interpretable features of the immunological synapse to predict the effectiveness of therapeutic antibodies on T cell cytokine production. These features allowed us to predict the functional outcome of an unseen antibody and to pinpoint the driving factors required for the prediction. For the TCBs, we found the intensity of MHCII and morphology of F-actin as the most prominent features in predicting cytokine readouts. These features can be used as inspiration for further investigation and formulation of new hypotheses on the mode of action of TCBs. Nonetheless, additional experiments are necessary to validate them. The ability to predict unseen antibodies could potentially enable the investigation of various antibody formats to understand mechanistically better how different formats can impact T cell responses and help to guide format selection.

scifAI is an end-to-end data acquisition and analysis framework which can be adjusted to investigate various hypotheses and to develop diverse applications based on imaging flow cytometry data. For instance, while in this study, memory CD4+ T cells were analyzed as they are poised to show faster immune responses and a higher synapse propensity compared to naive T cells[48], imaging and analysis of CD8+ T cells, as the main players in cytotoxicity, could further elaborate how synapse features correlate with killing efficiency of therapeutic antibodies against tumor cells. scifAI can also be utilized in the design of IFC experiments, optimizing the number and type of stainings, as well as the total number of images per donor to be acquired. In Pharma R&D, scifAI has the great potential to improve the quality and the speed of antibody development, for example, giving new insights into the mode of action of particular candidate molecules or predicting in vitro efficacy in high-throughput. AI-assisted identification of lead molecules and better prioritization in terms of epitope, affinity, avidity, and antibody format can greatly impact the decision-making process. In a nutshell, IFAI could provide substantial benefits by assisting in the investigation of the mode of action and the functionality of newly generated antibody candidates.

## Methods

### Human PBMC isolation
Blood samples from healthy donors were obtained through the internal Medical Service from Roche Diagnostics GmbH at Penzberg, Germany, with the approval of the ethics committee of the Bayerische Landesärztekammer. The donors were volunteers who gave informed consent for experimental research work and were selected based on availability and independent of age and gender by the Medical Service. An overview of the gender, number and age of participants for each experiment can be found in Supplementary Fig. 3a. No gender analysis and disaggregation according to sex were carried out due to insufficient numbers of individuals of different genders. PBMCs were isolated from whole blood by density-gradient centrifugation over Pancoll (density: 1077 g/mL, PAN-Biotech, cat # P04-601000).

### Cell line culture
EBV-transformed B-lymphoblastoid cell line (B-LCL) from donor 333 was obtained from Astarte Biologics (# 1038-3161JN16), and cells were cultivated in RPMI-1640 medium (PAN-Biotech; cat # P04-17500) with 10% FBS (Anprotec; cat # AC-SM-0014Hi) and 2 mM L-glutamine (PAN-Biotech; cat# P04-80100). Z138 (MCL, gift from University of Leicester) and Nalm-6 (ALL, DSMZ ACC 128) tumor cells were cultivated in RPMI1640 containing 10% FBS and 1% Glutamax (Invitrogen/Gibco # 35050-038).

### Immune synapse formation and imaging flow cytometry
To analyze immune synapses, human memory CD4+ T cells were isolated from peripheral blood mononuclear cells (PBMCs) of nine healthy human donors using a negative selection EasySep Enrichment kit from STEMCELL Technologies (cat #19157). Live/dead staining of T and B-LCL cells was separately performed using the fixable viability dye eF780 for 15 min at RT (eBioscience; cat # 65-0865-14). Cells were then re-suspended in RPMI-1640 medium supplemented with 10% FBS (Anprotec; cat # AC-SM-0014Hi), 5% Penicillin-Streptomycin (Gibco; cat # 15140-122), and 2 mM L-glutamine (PAN-Biotech; cat # P04-80100). Afterward, B-LCL cells were transferred into a well of a 96-well round bottom plate (300,000 cells per well) and were pre-incubated with the superantigen Staphylococcal enterotoxin A (SEA) (Sigma-Aldrich; cat # S9399) for 15 min at 37 °C or left untreated. Human CD4+ $T_{mem}$ were added to the afore-prepared B-LCL cells (250.000 cells per well) to generate a final ratio of 4:3 (B-LCL:$T_{mem}$), and subsequently, the appropriate in-house made compounds (10 µg/mL of Isotype Ctrl or Teplizumab and 1 µg/mL (5 nM) of Ctrl-TCB, CD19-TCB[49] or CD20-TCB[15,53]) were added to the B-LCL-$T_{mem}$ cell co-culture. To strengthen the conjugate formation between B-LCL and T cells they were centrifuged at 30×g for 30 s and then directly transferred to a 37 °C incubator for 45 min. Thereafter, the medium in each well was carefully aspirated with a pipette, and cells were immediately fixed for 12 min at RT followed by permeabilization using the Foxp3/Transcription factor staining buffer set from eBioscience (cat # 00-5523-00). Intracellular staining was performed in permeabilization buffer containing fluorescently labeled antibodies for 40 min at 4 °C: CD3-BV421 (clone UCHT1, Biolegend; cat # 300433; 1:20), HLA-DR-PE-Cy7 (clone L243, Biolegend; cat # 307616; 1:200), Phalloidin AF594 (ThermoFisher; cat # A12381; 1:600) and P-CD3ζ Y142-AF647 (clone K25-407.69, BD cat # 558489; 1:20). After washing, cells were suspended in FACS buffer (PBS supplemented with 2% FBS) and acquired on an Amnis ImageStream[X] Mark II Imaging Flow Cytometer (Luminex) equipped with five lasers (405, 488, 561, 592 and 640 nm). On average, around 55,000 images were collected per sample at 60x magnification on a low-speed setting. IDEAS software (version 6.2.187.0, EMD Millipore) was used for data analysis and labeling of cells. To identify immune synapses using the IDEAS software the gating strategy in Supplementary Fig. 1a was implemented. Cells were first gated on in-focus live+ CD3+ MHCII+ cells using the features area, aspect ratio, gradient RMS, and intensity of the respective fluorescent-labeled markers. Within this population images that show single CD3+ T cells and single MHCII+ B-LCL cells were selected using the area and aspect ratio feature. Next, to exclude non-interacting cells the CD3 intensity within a self-created synapse mask was determined. The synapse mask was defined as a combination of the morphology CD3 and MHCII mask with a dilation of 3. Only synapses that showed a CD3 signal in the mask were gated. Finally, T + B-LCL cells in one layer were excluded by using the height and area feature of the brightfield (BF) and single T-B-LCL synapses were analyzed. For each experiment, a compensation matrix was calculated to minimize spillovers into the different channels (see Supplementary Fig. 2).

## Conventional flow cytometry

For analysis of cell surface markers, live/dead staining was first performed using the fixable viability dye eF780 for 20 min at 4 °C (eBioscience; cat # 65-0865-14). Afterward, cells were pre-incubated with human Fc-block (BD, cat # 564220) in FACS buffer (PBS supplemented with 2% FBS and 1 mM EDTA) for 10 min at 4 °C and then stained with the appropriate fluorescently labeled antibodies for 30 min at 4 °C: CD4-BV510 (clone RPA-T4, Biolegend: cat # 300546; 1:100) or CD4-BV421 (clone RPA-T4, BD; cat # 562424; 1:50), HLA-DR-PE-Cy7 (clone L243, Biolegend; cat # 307616; 1:200), CD69-PE (clone FN50, Biolegend; cat # 310906; 1:100), CD80-APC (clone 2D10, Biolegend cat # 305220; 1:200) or CD86-PE (clone IT2.2, Biolegend cat # 305406; 1:200). For intracellular cytokine staining cells were first treated with GolgiPlug (BD Biosciences; cat # 555029) and GolgiStop (BD Biosciences; cat #554724) for at least 2–4 h before being stained. After incubation, live/dead staining was performed using the fixable viability dye eF780 for 20 min at 4 °C (eBioscience; cat # 65-0865-14). Cells were then fixed and permeabilized using the Foxp3/Transcription factor staining buffer set from eBioscience (cat # 00-5523-00) as described for the synapse formation assay. Intracellular staining was performed in permeabilization buffer containing the fluorescently labeled antibody Granzyme B-PE-Cy7 (clone QA16A02, Biolegend; cat # 372214; 1:50) or TNF-α (clone MAb11, BD; cat # 554514; 1:50) for 30 min at 4 °C. Finally, cells were suspended in FACS buffer (PBS supplemented with 2% FBS and 1 mM EDTA) and acquired on a FACS Celesta from BD Biosciences.

A representative example of the gating strategy used for analyzing conventional flow cytometry data in this study is shown in Supplementary Fig. 11. Briefly, lymphocytes were selected in the FSC-A and SSC-A gate. In the next step, single cells were selected using FSC-H/FSC-W, and viable cells were identified using the fixable viability dye eF780 (gated on eF780 negative cells). Finally, cells were gated on CD4$^+$ T cells, and markers of interest were analyzed (see Supplementary Figs. 1d and 10a).

## Tumor cell lysis assays (in vitro)

B cell-depleted PBMCs derived from the blood of healthy donors were prepared using standard density-gradient isolation followed by B cell depletion with CD20 Microbeads (Miltenyi; cat # 130-091-104). B cell-depleted PBMCs were then incubated with the tumor targets (Z-138 or Nalm-6) at a ratio of 5:1 for 24 h in the presence or absence of CD20-TCB or CD19-TCB. Tumor cell lysis was calculated based on LDH release (LDH Cytotoxicity Detection Kit from Roche Applied Science) and normalized to spontaneous release (PBMCs + targets without treatment = 0% tumor cell lysis) and maximal release (lysis of tumor targets with Triton X-100 = 100% lysis).

## Quantification of CD20 and CD19 expression

CD19 and CD20 expression on B-LCL cells were determined using the Quantum™ Alexa Fluor® 647 MESF Kit from Bangs Laboratories (cat # 647) according to the manufacturer's instructions using an anti-human CD20-AF647 (Biolegend # 302318) or an anti-human CD19-AF647 (Biolegend # 302220) antibody as well as the corresponding isotype controls muIgG1 (Biolegend # 400130) and muIgG2b (Biolegend # 400330). For the quantification of CD19 and CD20 molecules on the tumor target cell lines Nalm-6 and Z-138, the QiFi Kit from Dako (cat # K0078) was performed according to the manufacturer's instructions by using an anti-human CD20 purified (BD # 555621) or an anti-human CD19 purified (BD # 555410) antibody as well as the corresponding isotype controls muIgG1 (BD # 554121) and muIgG2b (BD # 557351).

## Preparation of the imaging dataset for analysis

We recorded 2,899,575 images from the commercial imaging flow cytometer, Luminex Amnis ImageStreamX Mark II Imaging Flow Cytometer, with estimated throughput of 100–200 events/s. The dataset consists of nine distinct donors across four independent experiments. Donor 1 and Donor 2 were used twice (Supplementary Fig. 3a). Different conditions were measured which included -SEA (total images=625,001), +SEA (557,781), Ctrl-TCB (330,000), CD19-TCB (324,020), CD20-TCB (254,398), Isotype (405,000), and Teplizumab (403,375). The images contained brightfield (BF), F-actin, MHCII, CD3, P-CD3ζ, and Live/Dead stainings. The Live/Dead staining is only used to filter out the dead cells. For each experiment, the images were compensated using a compensation matrix derived from stained single cells. After the compensation, the raw images (16-bit) and their corresponding channel-wise segmentation masks were exported from the IDEAS software and saved in an HDF5 format. To enable parallelization, each image and its corresponding mask were saved separately. In addition, our expert annotated a subset of data for -SEA (labeled images = 1160), +SEA (4061), CD19-TCB (396) and Teplizumab (227). The labeled ±SEA-based data (5221) was used for training and validation of the classification models, where 70% was used for training and 30% for testing in a stratified way.

## Interpretable feature engineering from images

We extracted a set of 296 biologically motivated features to study the immunological synapse. These features included morphology, intensity, co-localization, texture and synaptic-related values (see Supplementary Fig. 4). The morphology features were calculated based on the segmentation mask from each channel. The features included 'area', 'bounding box area', 'convex area', 'eccentricity', 'equivalent diameter', 'Euler number', 'extent', 'maximum Feret diameter', 'minimum Feret diameter', 'filled area,' 'length of major axis', 'length of minor axis', 'Hu moments', 'orientation', 'perimeter', 'Crofton perimeter', 'solidity', 'weighted Hu moments'. All the morphology features are extracted using scikit-image library[61]. For the intensity features, first the cells were segmented using their corresponding mask. The intensity features included 'min', 'sum', 'mean', 'standard deviation', 'skewness', 'kurtosis', 'max' and 'Shanon entropy'. In addition, the percentile of intensity values, including '10th percentile', '20th percentile', ..., '90th percentile' were calculated. All of the intensity features were calculated based on NumPy[62] and SciPy[63] functionality. For co-localization features, we implemented 'dice distance' and 'Jaccard distance' to calculate the masks overlap between two channels using the SciPy[63] library. In addition, we calculated the 'correlation distance'[63], 'Euclidean distance'[63], 'Manders overlap coefficient'[64], 'intensity correlation quotient'[64], 'structural similaity'[61] and 'Hausdorff distance'[61]. For texture features, we used Gray Level Co-occurrence Matrix (GLCM) features[65] including 'contrast', 'dissimilarity', 'homogeneity', 'ASM', 'energy' and 'correlation'. The synapse related features were defined as 'enrichment of Ch (mean)'=$mean(intensity\ of\ Ch\ in\ synapse)/mean(intensity\ of\ Ch)$, 'enrichment of Ch (sum)'= $sum(intensity\ of\ Ch\ in\ synapse)/sum(intensity\ of\ Ch)$, and 'enrichment of Ch (max)'=$max(intensity\ of\ Ch\ in\ synapse)/mean(intensity\ of\ Ch)$[33]. Finally, we implemented 'background mean' and 'gradient RMS' for quality control of images. All these features were implemented using NumPy (version=1.18.5), Pandas (1.1.5), SciPy (1.8.0), scikit-image (0.19.2), and scikit-learn(1.0.2)[66].

## Autoencoder feature extraction

To leverage the large amount of unlabeled data, we implemented and trained a multichannel autoencoder[25]. This autoencoder included a separate encoder for each channel. The encoders were designed to map each channel to a 32-dimensional vector. The concatenation of these vectors led to a 5*32 dimensional space. Then these features were mapped to a 256-dimensional feature vector. A decoder on top of the concatenated vectors was implemented for reconstructing the original image. Mean squared error (MSE) was used as the reconstruction loss. The augmentations used for training the autoencoder included random rotation, random scaling, random flipping, and random Gaussian noise.

### Feature pre-selection

Considering that the number of features was large, we implemented a feature pre-selection pipeline to select the most relevant features. We followed the work of Haq et al.[67] (see Supplementary Fig. 5a). First, the Pearson correlation between the features was measured. If at least two features were highly correlated (|corr| >0.95), then only one of them was kept (at random), and the rest were eliminated. Six different methods were used to rank the features in the next step. These methods included mutual information, linear support vector machine, logistic regression with L1 regularization, logistic regression with L2 regularization, random forest, and XGBoost. The *top-k* (hyper-parameter to be selected) features from each method were selected, and their union was used. After this reduction, the Spearman correlation matrix between the features was calculated, and spectral clustering was performed on the correlations. Then, $m$ clusters were created, and one feature at random per cluster was selected. The last step was performed to account for multicollinearity between the features.

### Classification

There are three main approaches that are used for training a supervised learning algorithm in this work, feature-based approach and deep learning.

### Classical supervised learning models

We used two different algorithms for training machine learning models. We used a boosting method called XGBoost[37] which uses an ensemble of trees on the data (n_trees = 100). The second model was logistic regression. The advantage of using these models was that they provide explainability after the training.

### Convolutional neural networks

For training supervised deep learning models, we used well-known architectures in the field of computer vision, including ResNet18, Resnet34, and DeepFlow[23–25]. All models were pre-trained on ImageNet. Considering these models are originally designed for three RGB channels input, we had to substitute the first convolutional layer with three input channels to five input channels, including BF, F-actin, MHCII, CD3, and P-CD3ζ. In addition, the classification layer also needed to be adjusted to have nine classes. Barlow Twins[47] was used for pre-training the ResNet18 and ResNet34 networks using the unlabeled data. Next, we used multi-class cross-entropy loss for training on the labeled data. The learning rate (lr) was set to 0.001, with the adaptive strategy of reducing on the plateau of 10 epochs. The augmentations used for training included random rotation, random scaling, random flipping, and random Gaussian noise. 15% of the training data was selected randomly and used as a development set to avoid overfitting.

### Classification feature importance

As mentioned, a feature pre-selection filtering was used to reduce the number of features, and then an XGBoost classifier was trained on the annotated data. While the XGBoost can provide in-model feature importance, these importances can be biased due to different reasons, such as the correlation between the pre-selected features, the number of features, the pre-selection process, outliers, etc. To account for this, we split the training data randomly to 5-fold (stratified) and trained the XGBoost classifier five times, each time using 4 out of the 5 folds. We repeated this process 100 times, leading to 500 different models. In each training, we used a random number of pre-selected features (top-k) between 30 to 200 features. Eventually, for every feature, a series of feature importances were obtained. The median feature importance for each feature was used to rank the features.

### Classification performance validation and generalizability

We validated the prediction performance of classification models across multiple datasets. To offer a comprehensive view of the generalizability of our results, we present a complete list of all the steps taken. The first validation was performed on the ±SEA-based test set (30% hold-out set) to confirm inter-experiment comparability (Supplementary Fig. 5c). The second validation was performed using leave-one-donor-out cross-validation to confirm inter-donor generalizability (Supplementary Fig. 6b). Next, the model was validated on two separate test sets with Teplizumab-based and CD19-TCB-based perturbations to validate inter-experiment and inter-stimulation generalizability (Supplementary Fig. 9). Overall, we demonstrated that the XGBoost model trained on ±SEA perturbations could indeed generalize to new perturbations with previously unseen antibodies and inter-donor variability.

### Feature importance of the XGBoost model

Average gain was used to determine the XGBoost model's feature importance. To calculate the average gain for a specific feature, it is necessary to find all trees and splits that used that feature. Then, the improvement by each split in the multi-class loss function is calculated. Finally, the average gain is calculated by summing all improvements and dividing this sum by the total number of splits involving the feature of interest.

### Classification staining importance

To determine which staining contributes the most to the predictions, we used recursive channel elimination. In every run, we always kept BF as it is stain-free. Then train the 'interpretable features + XGBoost' based on the features of the selected channels (see Supplementary Fig. 7b).

### Data cleaning pipeline

For each donor, we used the trained XGBoost classifier to predict the class of every image. We excluded low-quality and outlier images based on the following data cleaning protocol, where we first describe the types of cells we filtered out, followed by the concrete feature-based rule:

1. Dead cells (using Live/Dead staining): 'mean Live/Dead intensity' >= 'mean Live/Dead intensity (90th perc.)'
2. Out-of-focused images: 'Gradient RMS BF' > 'Gradient RMS BF (2nd perc.)' & 'Gradient RMS BF' <'Gradient RMS BF (90th perc.)'
3. High entropy images, based on the XGBoost predictions: entropy > 1.0. The entropy was calculated using the SciPy package. This step is done to omit images that the classifier is the most uncertain in terms of prediction.
4. Images predicted as 'B-LCL' with low MHCII: 'mean intensity of MHCII' <'mean intensity of MHCII (5th perc.)'. This step guarantees that the images predicted as 'B-LCL' contain a minimum of MHCII intensity.
5. 'B-LCL' with 'area of MHCII' <'area of MHCII (10th perc.)'. This step guarantees that the images predicted 'B-LCL' contain a cell with appropriate size.
6. 'T cell' with 'mean intensity of CD3' <'mean intensity of CD3 (1st perc.)'. This step guarantees that images predicted as 'T cell' contain a minimum CD3 intensity.
7. 'B-LCL and T cell in one layer' with small 'B-LCL's: 'B-LCL and T cell in one layer' with 'area of MHCII' <'area of MHCII (20th perc.)'. This step is performed to omit misclassified 'B-LCL and T cell in one layer' with small 'B-LCL's.
8. Outlier detected via isolation forest[68]: We used n_estimators = 100, max_samples = 'auto', contamination = 'auto', and max_features = 20 as the main parameters. We only used the top 30 features based on feature importance from the XGBoost training to reduce the run time.
9. Outlier detected via DBSCAN[69]: First, we transformed all images to 2D dimensional space using Uniform Manifold Approximation and Projection (UMAP). The features were standardized using the

mean and std of each feature. For reducing the run time, we only used top 30 features based on feature importance from the XGBoost training. Then a DBSCAN algorithm was run with eps = 0.09 and min_samples = 5. The resulting clusters were filtered out if (#images in cluster)/(#total images) <0.0001.

All these steps are based on scikit-learn implementations. All parameters were set using the default value of scikit-learn unless stated otherwise. Twenty random examples of filtered-out images are shown in Supplementary Fig. 8.

### Class frequency analysis

After the data cleaning, the frequency of each class was calculated with 'F_C = (#images predicted as C)/(#total images)' for each condition per donor. To deal with the compositional nature of the data, we used $\log_2(F\_C\_antibody/F\_C\_control)$ to compare the frequency fold changes. This transformation has the advantage that the frequencies do not sum to a constant value. After calculating the $\log_2$ fold changes, we used the Wilcoxon-rank-sum test to analyze the effects of antibodies on class frequencies. Wilcoxon-rank-sum tests whether two samples are likely to derive from the same population. To account for multiple testing, we used Benjamini-Hochberg correction for +SEA/-SEA, CD19-TCB/Control TCB and Teplizumab/isotype, respectively. Because the experiments were performed independently, we only corrected each of these comparisons separately.

### Feature difference analysis

We analyzed the effect of perturbation with CD19-TCB, and Teplizumab on signaling synapses using the previously trained XGBoost model from the ±SEA training set and then used statistical tests to investigate the mode of action of these therapeutic antibodies. First, we selected the images predicted as 'synapse w/ signaling.' We only focused on fluorescent channels as they contain targeted information on components of the cell expected to morphologically change during synapse formation. This procedure yielded 210 features for comparison for TCB based on F-actin, MHCII, CD3, and P-CD3ζ. For Teplizumab, CD3 was not available for the analysis because of the usage of CD4 in recording images for Teplizumab instead of CD3. Thus we analyzed 132 features extracted from F-actin, MHCII, and P-CD3ζ. After the feature selection, we compared the features using the Mann-Whitney U test for each condition and its control. To understand the direction of change, we used the difference in the median of features for each condition and its control. To account for multiple testing, we used the Benjamini-Hochberg procedure with α = 0.05. As the conditions were independent, we corrected the p-values for each condition and its control separately. If the test was not significant, we assigned that feature 0 (gray in Fig. 3a, h and 0 in Supplementary Tables 1 and 2). If it was significant and the median value of the feature for the antibody was greater than the median of the feature of the control antibody, we assigned +1 (red in Fig. 3a, h and +1 in Supplementary Tables 1 and 2). On the contrary, if the test was significant, but the median value of the feature for the antibody was smaller than the median value of the feature of the control, we assigned −1 (blue in Figs. 3a, h and −1 in Supplementary Tables 1 and 2).

### Validation of the biological findings

In the study of the mode of action of antibodies, six donors for the CD19-TCB and seven donors for Teplizumab were analyzed, respectively. To avoid systematic bias in the analysis due to potential batch effects and emphasize the generalizability of the results, we included at least two independent experiments for each antibody: experiments III and IV for CD19-TCB and I, II, and III for Teplizumab. This allowed us to focus on consistent findings between the experiments. All statistical tests were corrected for multiple testing to account for the inflated occurrence of false positive results. Finally, visual inspections of the acquired images were done by expert immunologists, and the findings were validated in the literature.

### GrzmB prediction and feature ranking

To predict GrzmB, we only used images predicted as 'synapse w/o signaling' and 'synapse w/ signaling' for each condition. This choice was made as it is assumed that the synapses will lead to cytokine production. Considering that we had thousands of images for each donor and condition, we used an aggregation pipeline to create a feature vector corresponding to each donor and condition. To reduce the number of features, we only used the consistent feature changes for the CD19-TCB (Fig. 3). For each donor and condition, we aggregated the features using the 5th, 50th, and 95th percentile to capture the extremes and average of every feature.

After deriving the aggregated features, we used one-donor-leave-out cross-validation to train a linear regression model with LassoLars. The most important features were based on the magnitude of the coefficients.

### Visualizations and tables

For visualizing the conjugates and biological context, we used www.BioRender.com. We used matplotlib (version 3.3.2) and seaborn (0.11.2) in Python for the plots and images. Finally, Inkscape was used for creating the figures and tables, including the glossary of all the abbreviations in Supplementary Table 3.

### Statistics and reproducibility

For reproducing the results or running exemplary code provided as part of the software package, it is necessary to download the dataset and install scifAI. Extensive documentation has been provided that will allow users with basic programming knowledge to follow four application examples and, with minor modifications, adapt the code to their needs. To tackle more advanced use cases, such as defining a new feature or applying scifAI to an entirely new dataset, more programming experience is required.

### Reporting summary

Further information on research design is available in the Nature Portfolio Reporting Summary linked to this article.

## Data availability

All imaging flow cytometry data generated in this study, including all data and instructions to reproduce all findings described in this article and the supplementary information, have been published and can be freely downloaded at Datadryad[70] under the following link: https://doi.org/10.5061/dryad.ht76hdrk7. Additional metadata can be requested from the corresponding authors upon request. Source data are provided as a Source Data file. Source data can also be found in this repository https://github.com/marrlab/scifAI-notebooks. Source data are provided with this paper.

## Code availability

The scifAI code and instructional notebooks on how to run the code and build analysis pipelines can be found at: https://github.com/marrlab/scifAI/. https://github.com/marrlab/scifAI-notebooks. For reproducing the results, a minimal dataset is provided in the repository.

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

## Acknowledgements

S.S.B. has received funding from F. Hoffmann-la Roche LTD (No grant number is applicable) and is supported by the Helmholtz Association under the joint research school 'Munich School for Data Science (MUDS)'. The authors thank Ronan Le Gleut from the Core Facility Statistical Consulting at Helmholtz Munich for the statistical support. We thank Florian Limani from the Roche Innovation Center Zurich for running the killing assays. We acknowledge the Core Facility Flow Cytometry at the Biomedical Center, Ludwig-Maxmilians-Universität München, for providing equipment, services, and expertise. We also thank Martin Turner for the fruitful discussions at the beginning of the project that challenged the biological combination of cell markers. We thank all members of the CD19-TCB and CD20-TCB teams as well as colleagues from Large Molecule Research (LMR), especially Alain Tissot, Diana Pippig, and Olaf Mundigl, and colleagues from Cancer Immunotherapy, Oncology at Roche Pharma Research and Early Development (pRED) for their valuable input and scientific discussion. C.M. acknowledges funding from the European Research Council (ERC), grant agreement no. 866411, and support from the Hightech Agenda Bayern. Finally, we thank Tijana Nikic for their support in the annotation process.

## Author contributions

K.E. designed and performed the experiments. S.S.B. wrote the code, analyzed the data and created the figures. S.S.B. and K.E. wrote the manuscript with input from E.G., C.M., and F.S. E.G., C.M., and F.S. conceived the study. N.K. and F.T. helped with the manuscript narrative, analysis of ideas, and editing. S.H. and M.B. provided scientific input, designed experiments, and supported the writing of the manuscript. All the authors have read and approved the manuscript.

## Competing interests

The authors declare no competing interests.
