## [Peer Review File · Nature Communications]

Reviewers' Comments:

Reviewer #1:

Remarks to the Author:

This manuscript proposes a learning method with feature interpretability for revealing underlying relationships between immunological synapses and functional characterizations of therapeutic antibodies. In general, I think this work is clinically meaningful, and the manuscript is well organized. However, I do not think this work is worth being published in Nature Communications in its current form:

1. The authors mention over 2.8 million images were acquired, while how this large amount of data was used and contributed to the results is missing.
2. Did the authors use a commercial imaging flow cytometer or they constructed their cytometer on their own? What is the throughput of the cytometer? Because high throughput is a key factor of this work.
3. How the algorithm supports high-throughput data processing should be explained.
4. "Synaptic features were implemented based on the ratio of the signal intensity of each fluorescent channel in the synaptic area to the whole cell." How was the ratio determined?
5. There should be a validation set when training the algorithm.
6. Why is it necessary to evaluate the relationship between classification performance and the number of annotated samples? How should we understand the finding that "We found that the channels BF, MHCII and P-CD3 ζ sufficed to reach an F1-macro similar to using all the channels (Supplementary Fig. 5b)"?
7. L2 norm is not usually used as a loss function in machine learning, why did the authors choose it?
8. Why XGBoost and logistic regression can improve explainability, and what does explainability mean in detail?
9. "Considering the models are designed for three channels input, we had to remove the first convolutional layer with three input channels to six input channels." Why, and what are the six inputs channels?
10. Although the total number of cellular images is large, how many donors were involved in the experiment? It seems the number is not that large.
11. There are also some typos:
 - a. The values of the scale bars in supplementary Fig. 3 are missing;
 - b. There is no panel c in supplementary Fig. 5;
 - c. There is no panel e in Fig. 1;

Reviewer #2:

Remarks to the Author:

In this article, Boushehri & Essig et al. presented an open-source machine learning-based analysis framework for single-cell imaging flow cytometry, as well as a public release of an annotated immunological cellular image dataset. The overall work is scientifically sound and relevant to both the biomedical and data science fields, with appropriate methodologies and analytic approaches. However, major improvements are needed to meet the publishable expectations.

Major comments:

* In the first major result "comprehensive multi-channel IFC dataset", please reassess and/or elaborate on why it is admirable to achieve such a dataset.

If it is about the volume ("largest-to-date"), please indicate the multitudes of challenges in wet-lab/dry-lab to reach such a volume, if there are any. IFC is known for its high-throughput image acquisition rate at thousands of events per second; obtaining millions of images should not be difficult.

If it is about the annotation/labeling effort for ground truth, the main contributor for the annotation seems to be a single person, K.E. as described in several places in the article. While we could much appreciate the hard work and the good intention of promoting open science, please explain the unique expertise of this annotator that would make such a dataset truly close to ground truth.

If it is about the variety of the dataset, please justify why six-donor inclusion in this study is a good-enough sample size, representative and generalizable for a larger population.

* As noted above regarding the limited number of annotator experts: please conduct a measurement for interobserver and intraobserver variability. Without such evidence, it is very difficult to appreciate a high level of fidelity for the dataset, and the subsequent controls for overfitting during model training/testing.

* Explainable AI is not the same as having a list of human-readable features. Whether the authors agree or disagree with that statement, please provide supporting evidence and/or discussion. In part two and part three of the "Results" session, "sciAI: An explainable AI python framework..." and "sciAI enables high-throughput profiling...", what stood out was: the authors implemented and benchmarked an XGBoost classifier that used pre-engineered handcrafted features, against other deep learning-based classifiers that contained difficult-to-explain representation embeddings. That work alone would not result in interpretability. For example:

- A single feature may have a clear name but does not have a clear meaning in biology: what does it mean when there are big or small cells in an immune reaction? Are big cells activated T cells or dividing cells or inactivated cells?

- A combination of meaningful features might result in better machine learning classification performance but might not have a clear biological reason. For example, big sizes and rough textures might be meaningful when they stand alone; but what does it mean the combination of both, is it a good or bad immune phenotype?

The most useful discussion of the interpretability was presented at the end of part 3 of the "Results" session, where authors noted the most predictive features were co-localization, intensities, and textures of biomarkers (CD3, MHCII, P-CD3ζ etc.). However, please provide further meaningful evidence of why the high/low of Kurtosis, Skewness, Contrast, and Correlation distance are good proxies of biological processes; e.g. are those high/low intensity-based features (and/or their combinatorial effects) equivalents of increase/decrease membrane protein expressions?

* The authors first noted "channels BF, MHCII and P-CD3ζ sufficed to reach an F1-macro...", but later in the part "Characterizing the impact of therapeutic antibodies on synapse formation" indicated that "[BF] intensity is difficult to interpret and its morphological characteristics can be captured also by other fluorescent channels". This is contradictory:

- How could an explainable AI framework have room for the exclusion of difficult-to-interpret features?

- Secondly, if BF could be easily replaceable, why would the author praise its usefulness in the first instance?

* The results of CD19-TCB and Teplizumab on synapse formation, with its associated Figure 3 and discussion were great. Appreciate the detailed presentation.

* Exploring the use of image features in predicting cytokine production is ambitious, and it is unconvincing when the number of samples is low, as noted by the authors themselves. While the attempts are appreciated, please further prove the alignment of the "standard deviation of MHCII (95th perc.) [intensity]" and "eccentricity of F-actin (95th perc.) [morphology]" with the biological process of cytokine production.

Minors:

* Please spell out the first instances of abbreviations, or make an effort for a glossary table, such as "MHC", "EBV", "FACS", "PBMCs" etc.

* There was an inconsistency in how the manuscript writers indicate cellular conjugates: sometimes "T:APC", other times "T-B"

* Figure preparations were great, but figure captions were not, for example, Figure 2 captions are inadequately described. "a, c Schematic representation of the mod of action ...", please describe those modes of action. Similar expansions to supplementary figures' captions are also needed.

* In the "Introduction" session, second paragraph, last sentence "IFC has recently been successfully ... the immunological synapse of ...T:APC cell, however, none of these investigated ... the immunological synapse in the context of T cell function". This important sentence is confusing, what is the actual difference between the first and the second mentioning of "immunological synapse".

Reviewer #3:

Remarks to the Author:

The work establishes an open access software scifAI based on imaging flow cytometry and explainable machine learning to predict the functionality of therapeutic antibodies in vitro. It also includes 2.8 million IFC cell images of human immunological synapse and makes these images available to the public. This work uses interpretable features of the immunological synapse to predict the effectiveness of therapeutic antibodies on T cell cytokine production to allow prediction of the functional outcome of an unseen antibody. There are many positive points, making the paper a worthy addition to the scientific literature. However, several major issues need to be addressed before acceptance of the work for publication:

1. The authors did not discuss whether the cell images and the explainable AI algorithm are dependent on the image modalities or specific brand/model of IFC. Can the image data be used to train cell images from different imaging equipment such as a confocal microscope or different models of imaging flow cytometer? Can these images in the data base be used to train cell images with different magnification factor, wavelengths, or focal depth? Basically, the authors need to define and specify any constraints of the usability of the their data.
2. The authors mention that they throw away some images for different reasons (out of focus, doublets, blur, etc.). It would be helpful for readers to understand what images are "acceptable" and what should be 'rejected'. Is there a set of quantitative criteria to determine the qualified images for the machine learning model?
3. The paper put great emphasis on "explainability" of the AI results. Explainable AI is obviously the key component of the paper. However, it is not clear to the reviewer how the "explainability" is measured and verified. The paper shows agreement between human recognizable image features (e.g. co-localization) and AI results, but the co-relation between the two does not necessarily indicate that this is how AI uses to produce its results. There is no analysis such as CAM to better assure or elucidate AI explainability.

Reviewer #4:

Remarks to the Author:

The authors generate an imaging flow cytometry dataset of B-T cell conjugates using CD4 T cells, B lymphoblastoid cells and superantigen and develop a machine learning-based method for the classification of the images to detect the formation of immunological synapses and extract biological information, in order to evaluate the function of different therapeutic agents. The work is

interesting and novel. However, it is not clear how generalizable it is for the work of other users, with different types of samples and/or biological questions. Also, it might be challenging to use for biologists without programming knowledge, therefore limiting the potential impact of the method. Major concerns include:

- Authors use a stratified sampling scheme which guarantees that the training set will be representative of all donors used for testing. It seems that training was not limited to the initial SEA driven synapse datasets, but was continuous with mention of keeping some types of data in the "training set". Does the model generalize for donors that are not part of the training set? Authors should clarify that the model is transferable between different donors by using different donors for training than for testing.

- Similarly, the problem dataset and the training dataset are part of the same bigger set of images and this could cause overfitting of the model. How robust is the model for datasets more different to the training set?

- How robust is it to day to day differences in instrument settings and have data subjected to analysis been collected on different machines. For example, does the model still work if using different image acquisition parameters (for example different laser power) or different types of samples, for example different subsets of B and T cells (like naive T cells (smaller) or blasted T cells (bigger))?

- It is impressive that the AI was able to correctly score both activation and inhibition of synapse formation and function. To better understand the outputs, is it correct that the scoring is all 0 or 1, no or yes, and the continuous data in, for example, Fig 3b, d, f, etc is an average of the binary results. Did the training consist of making these binary judgements or of indicating where to make quantitative measurements? As a biologist, I would like to take advantage of such information and my impression is that the more automated feature selection was performing some quantitative measurements with %tiled measurements, but were these measurement (such as Fig 3 a, h) only made after the classification using the human training data into the different major categories? I would appreciate if this could be clarified.

Minor:

- Fig 4a: legend mentions colours that are not present in the image.
- Suppl Fig 5: legend includes more panels than showed in the figure.

REVIEWER COMMENTS

Reviewer #1 (Remarks to the Author):

This manuscript proposes a learning method with feature interpretability for revealing underlying relationships between immunological synapses and functional characterizations of therapeutic antibodies. In general, I think this work is clinically meaningful, and the manuscript is well organized. However, I do not think this work is worth being published in Nature Communications in its current form:

1. The authors mention over 2.8 million images were acquired, while how this large amount of data was used and contributed to the results is missing.

We would like to thank the reviewer for raising this point. We have added the following sentences to the Results section of our manuscript to clarify the use of the acquired image data and how the sample size contributed to the results:

Formation of T cell immune synapses occurs at variable and relatively low frequencies depending on the donor, the APC, and pharmacological perturbations ^(9,12,19–21). Therefore, high-throughput IFC was selected as the method of choice to capture a large number of samples, enabling the detection of subtle changes in cell morphology. Using IFC we generated a comprehensive data set for the systematic analysis of the immunological synapse of T-B conjugates (Fig. 1a and Supplementary Fig. 1a).

We also elaborated on this in the Discussion:

We generated the largest publicly available imaging flow cytometry data with over 2.8 million images using human primary immune cells from nine donors in four independent experiments that were treated with various therapeutic antibodies to study synapse formation. The large number of acquired images across multiple experiments provided sufficient statistical power to enable the study synapse formation, a potentially low-probability event, going beyond previous works that did, for example, not consider inter-experiments effects or inter- and intra- donor variability ^{21,40}.

2. Did the authors use a commercial imaging flow cytometer or they constructed their cytometer on their own? What is the throughput of the cytometer? Because high throughput is a key factor of this work.

Following the reviewer's question, we now clarify this point in the Methods section of our revised manuscript:

We recorded 2,899,575 images from the commercial imaging flow cytometer, Luminex Amnis ImageStreamX Mark II Imaging Flow Cytometer, with estimated throughput of 100-200 events/sec.

3. How the algorithm supports high-throughput data processing should be explained.

We thank the reviewer for this suggestion and revised the manuscript accordingly. In the Results, we added a sentence to elaborate on this:

Considering the computational complexity of machine learning modeling on high-throughput data, all mentioned algorithms were implemented using parallel computing, fully leveraging high-performance computing (HPC) infrastructure to speed up the calculations by distributing the computational load on multiple CPUs (Methods).

Also, we picked this up again in the Discussion to provide more context:

To detect and study immunological synapses, we implemented an interpretable feature extraction and machine learning framework in python by only using well-maintained python modules. scifAI natively implements parallel computing, fully leveraging modern HPC infrastructure and allowing for efficient processing and analysis of high-throughput data by parallelizing tasks such as data preprocessing and model fitting across multiple CPUs. These choices guarantee performance, scalability, and reproducibility and facilitate the deployment into existing workflows, which differs from previous works that use a combination of CellProfiler, R and python for each stage of the analysis ^{27,44,45}.

4. "Synaptic features were implemented based on the ratio of the signal intensity of each fluorescent channel in the synaptic area to the whole cell." How was the ratio determined?

The definition of these features is based on the work of Wabnitz et al. (Wabnitz G, Kirchgessner H, Samstag Y. Qualitative and Quantitative Analysis of the Immune Synapse in the Human System Using Imaging Flow Cytometry. J Vis Exp. 2019 Jan 7;(143). doi: 10.3791/55345. PMID: 30663655) and was provided already in the Methods section.

To clarify this more, we added Supplementary Figure 4c, explicitly showing the definition of the synaptic features.

c Visual representation of how synaptic features are calculated. To calculate the enrichment of intensity in the synaptic area, the ratio of intersection over the union of cells was used. The intersection is calculated using the masks from CD3 and MHCII channels. For the sum-based enrichment feature, the sum of intensities in each region is calculated. The same logic applied to the mean-based and max-based enrichment features.

5. *There should be a validation set when training the algorithm.*

We thank the reviewer for this important remark. Throughout our work, we have validated our results rigorously, but obviously failed to report this appropriately in the manuscript. In the revised version of our manuscript, we now clearly defined our validation sets as \pm SEA-based, CD19-TCB-based, and Teplizumab-based. We added a paragraph in the Methods section for more clarity on this important point:

Classification performance validation and generalizability

We validated the prediction performance of classification models across multiple datasets. To offer a comprehensive view of the generalizability of our results, we present a complete list of all the steps taken. The first validation was performed on the \pm SEA-based test set (30% hold-out set) to confirm inter-experiment comparability (Supplementary Fig. 4c). The second validation was performed using leave-one-donor-out cross-validation to confirm inter-donor generalizability (Supplementary Fig. 4d). Next, the model was validated on two separate test sets with Teplizumab-based and CD19-TCB-based perturbations to validate inter-experiment and inter-stimulation generalizability (Supplementary Fig. 7). Overall, we demonstrated that the XGBoost model trained on \pm SEA perturbations could indeed generalize to new perturbations with previously unseen antibodies and inter-donor variability.

6. *Why is it necessary to evaluate the relationship between classification performance and the number of annotated samples? How should we understand the finding that "We found that the channels BF, MHCII and P-CD3 ζ sufficed to reach an F1-macro similar to using all the channels (Supplementary Fig. 5b)"?*

We want to thank the reviewer for this question. To clarify the raised points, we have modified the manuscript as follows:

Considering that manual annotation of images can be time-consuming, we performed an ablation study to investigate how many annotated samples were necessary to achieve a high classification performance. We repeatedly trained the model on stratified subsets of the \pm SEA-based training data (5%, 15%, ..., 95%) and evaluated the F1-macro on the \pm SEA-based test set. The results showed that by using 1500 images (45% of the training data), we could achieve 0.90 ± 0.01 F1-macro on the test set (Supplementary Fig. 5a). This result demonstrated that it is possible to halve the manual annotation time and still achieve a similar quality of classification performance, as compared to using the whole annotated set (0.92 ± 0.01).

Next, we investigated the effects of fluorescent channels on the classification performance to determine which antibodies are essential for the detection of synapses and which ones can be freely exchanged depending on the biological context. Considering that BF is a stain-free channel provided for free by IFC, we kept the BF channel fixed and added all possible combinations of the fluorescent channels to train the model. We found that the combination of the channels BF, MHCII, and P-CD3 ζ sufficed to reach an F1-macro of 0.91 ± 0.01 (Supplementary Fig. 5b), similar to using all the channels (0.92 ± 0.01).

7. L2 norm is not usually used as a loss function in machine learning, why did the authors choose it?

We thank the reviewer for this remark and apologize for the ambiguity. The correct term for the autoencoder loss is Mean Squared Error (MSE) [1-2]. This choice makes sense as the autoencoder is trained to predict pixel values, which can be approximated by continuous values. In the new version, we have corrected this:

“... mean squared error (MSE) ~~‘L2 norm’~~ was used as the reconstruction loss.”

[1] Hastie, T., Tibshirani, R., & Friedman, J. H. (2009). The elements of statistical learning: data mining, inference, and prediction. 2nd ed. New York, Springer.

[2] Kinalis, S., Nielsen, F.C., Winther, O. et al. (2019). Deconvolution of autoencoders to learn biological regulatory modules from single cell mRNA sequencing data. BMC Bioinformatics 20, 379.

8. Why XGBoost and logistic regression can improve explainability, and what does explainability mean in detail?

We thank the reviewer for this remark. We now clearly define the type of explainability we refer to in this work in the Results section:

scifAI's explainable design follows the definition of Singh et al.²⁹ and Tjoa et al.³⁰, where machine learning models are explained by either in-model mechanisms, such as Gini-index, or by using post-model methods, such as saliency maps³¹ for deep learning models. Using interpretable features with a model whose internal mechanisms can be readily analyzed, scifAI natively provides in-model explainability and is fully compatible with other libraries, such as SHAP³² or Captum³³, to provide further post-model explainability.

Additionally, we explained how a model such as XGBoost can help improve explainability and why it is crucial for our work in the Discussion in more detail:

In this work, we followed the definition of Singh et al.²⁹ and Tjoa et al.³⁰ in the context of the explainability of machine learning and AI. We regard the biologically motivated features as interpretable as they are meaningful and can potentially hint at underlying biological mechanisms. We also considered our XGBoost model explainable as it natively provides a feature importance measure³². This design also enables a deeper understanding of the model's decision-making using additional methods, such as SHAP³². The insights from the interpretable features and explainable models have the potential to generate new biological hypotheses, which can lead to a better understanding of the underlying mechanisms at play. It is essential to mention that explainable models can offer a helpful intuition only if trained on interpretable features and will fail to provide a meaningful interpretation when trained on abstract features such as the bottleneck layer of an autoencoder. The combination of interpretable features and explainable machine learning enabled us to identify various relevant classes, such as immunological synapses, with state-of-the-art accuracy. It also allowed us to investigate the morphological profiles of the immunological synapse in an unbiased way and characterize the mode of action of antibodies in a biologically relevant context. This methodology is thus a substantial contribution to the field, which *has so far* primarily focused on performance over interpretability by using ResNet CNN architecture as the backbone^{26,41}.

9. *“Considering the models are designed for three channels input, we had to remove the first convolutional layer with three input channels to six input channels.” Why, and what are the six inputs channels?*

Thanks for bringing this mistake to our attention. We have corrected the number of channels accordingly and added more information about the change in the network architecture in the Methods section:

All models were pre-trained on ImageNet. Considering these models are *originally* designed for three *RGB* channels input, we had to *substitute* the first convolutional layer with three input channels to *five input* channels including *BF, F-actin, MHCII, CD3 and P-CD3ζ*.

10. *Although the total number of cellular images is large, how many donors were involved in the experiment? It seems the number is not that large.*

We clarified the number of donors involved by adding the following paragraph to the revised Methods section:

Validation of the biological findings

In the study of the mode of action of antibodies, six donors for the CD19-TCB and seven donors for Teplizumab were analyzed, respectively. To avoid systematic bias in the analysis due to potential batch effects and emphasize the generalizability of the results, we included at least two independent experiments for each antibody: experiments III and IV for CD19-TCB and I, II, and III for Teplizumab. This allowed us to focus on consistent findings between the experiments. All statistical tests were corrected for multiple testing to account for the inflated occurrence of false positive results. Finally, visual inspections of the acquired images were done by expert immunologists, and the findings were validated in the literature.

We also addressed this in the discussion:

We generated the largest publicly available imaging flow cytometry data with over 2.8 million images using human primary immune cells from nine donors in four independent experiments that were treated with various therapeutic antibodies to study synapse formation. *The large number of acquired images across multiple experiments provided sufficient statistical power to enable the study synapse formation, a potentially low-probability event, going beyond previous works that did, for example, not consider inter-experiments effects or inter- and intra- donor variability*^{21,40}.

11. *There are also some typos:*

- a. The values of the scale bars in supplementary Fig. 3 are missing;*
- b. There is no panel c in supplementary Fig. 5;*
- c. There is no panel e in Fig. 1;*

We thank the reviewer for bringing these points to our attention. We revised the figures accordingly.

Reviewer #2 (Remarks to the Author):

In this article, Boushehri & Essig et al. presented an open-source machine learning-based analysis framework for single-cell imaging flow cytometry, as well as a public release of an annotated immunological cellular image dataset. The overall work is scientifically sound and relevant to both the biomedical and data science fields, with appropriate methodologies and analytic approaches. However, major improvements are needed to meet the publishable expectations.

Major comments:

** In the first major result "comprehensive multi-channel IFC dataset", please reassess and/or elaborate on why it is admirable to achieve such a dataset.*

We thank the reviewer for assessing our work and are happy to elaborate on the size of our dataset. In the revised version of the our manuscript, we now start the Results section with:

Formation of T cell immune synapses occurs at variable and relatively low frequencies depending on the donor, the APC, and pharmacological perturbations ^(9,12,19-21). Therefore, high-throughput IFC was selected as the method of choice to capture a large number of samples, enabling the detection of subtle changes in cell morphology.

If it is about the volume ("largest-to-date"), please indicate the multitudes of challenges in wet-lab/dry-lab to reach such a volume, if there are any. IFC is known for its high-throughput image acquisition rate at thousands of events per second; obtaining millions of images should not be difficult.

We thank the reviewer for this critical point. We fully agree that IFC is known as high-throughput, and generating the data is not the biggest hurdle. Instead, we want to emphasize that this dataset enabled us to do the first coherent functional study using large image data sets and the consequential analytical part to visualize, understand, and study synapse formation and link predictive features. Moreover, this dataset provides a valuable resource for future applications such as studying the mode of action using deep generative models.

To address these points, we have modified the Discussion accordingly:

In the present work, we established scifAI, a pipeline based on two innovative technologies, imaging flow cytometry, and explainable machine learning to understand the mode of action and predict the functionality of therapeutic antibodies *in vitro*. We analyzed morphological profiles of the immunological synapse to better characterize the mode of action of therapeutic antibodies early after the initiation of an immune response and to apply it to forecast downstream T cell responses. **This work is the first coherent**

functional study using large image data sets and the consequential analytical part to visualize, understand and study synapse formation and its link to predictive features.

We generated the largest publicly available imaging flow cytometry data with over 2.8 million images using human primary immune cells from nine donors in four independent experiments that were treated with various therapeutic antibodies to study synapse formation. The large number of acquired images across multiple experiments provided sufficient statistical power to enable the study synapse formation, a potentially low-probability event, going beyond previous works that did, for example, not consider inter-experiments effects or inter- and intra- donor variability^{21,40}. [...] However, exploiting transfer learning, self-supervised pre-training⁴¹, and domain adaptation and generalization techniques⁴², this dataset will be a valuable resource for future applications, such as transfer to other fluorescent imaging modalities, for example, confocal microscopy, where data can be scarce. Training deep generative models to understand the mode of action of therapeutic antibodies⁴³ is another exciting avenue for future research.

If it is about the annotation/labeling effort for ground truth, the main contributor for the annotation seems to be a single person, K.E. as described in several places in the article. While we could much appreciate the hard work and the good intention of promoting open science, please explain the unique expertise of this annotator that would make such a dataset truly close to ground truth.

Thank you for the appreciation of our hard work in annotating the data. We decided to use the expertise of K.E. for the annotation as she has a strong background in immunology and extensive knowledge in T cell biology. To investigate the validity of our ground truth, two new annotators and K.E. additionally labeled a subset of previously annotated data, allowing us to measure inter- and intra-rater comparability quantitatively. The following paragraph has been added to the manuscript:

To assess intra- and interrater variability, 100 labeled samples were randomly selected. Three annotators with diverse backgrounds (K.E. - immunology and T cell biology, S.S.B - data analysis, N.K.C. - IFC analysis) annotated the images, yielding Cohen's kappa²⁸ scores of 0.84 (intra-), 0.79 (inter-), and 0.66 (inter-rater), respectively with respect to the original annotation of the data, which gave us confidence in the reproducibility and validity of the original annotation data.

If it is about the variety of the dataset, please justify why six-donor inclusion in this study is a good-enough sample size, representative and generalizable for a larger population.

We would like to thank the reviewer for bringing up this important point. We would first like to clarify that, in total, we used nine donors in our study. While this is not enough to represent a large population with diverse immunological responses, it allows us to find consistent patterns across donors, going beyond the state-of-the-art. With respect to the generalizability of our findings, we extended the analysis and description in the following way:

i. We analyzed inter-donor generalizability by performing a leave-one-donor-out cross-validation and added the following sentence in the Results section:

To confirm the validity of the results, a leave-one-donor-out cross-validation using the interpretable features and XGBoost was performed. The cross-validation yielded F1-macro values of 0.88 ± 0.04 , demonstrating good generalizability across donors (Supplementary Fig. 4d).

ii. Classifier generalizability: We summarized all the validation steps performed throughout our work, which ensure that the XGBoost model is generalizable to new perturbations, antibodies, and donors. We added a new paragraph in the Methods section to cover all these steps:

Classification performance validation and generalizability

We validated the prediction performance of classification models across multiple datasets. To offer a comprehensive view of the generalizability of our results, we present a complete list of all the steps taken. The first validation was performed on the \pm SEA-based test set (30% hold-out set) to confirm inter-experiment comparability (Supplementary Fig. 4c). The second validation was performed using leave-one-donor-out cross-validation to confirm inter-donor generalizability (Supplementary Fig. 4d). Next, the model was validated on two separate test sets with Teplizumab-based and CD19-TCB-based perturbations to validate inter-experiment and inter-stimulation generalizability (Supplementary Fig. 7). Overall, we demonstrated that the XGBoost model trained on \pm SEA perturbations could indeed generalize to new perturbations with previously unseen antibodies and inter-donor variability.

iii. Generalizability of the biological findings: We included at least two independent experiments (batches) with different donors for each antibody. This helped us detect the differences induced by the antibodies and not potential batch effects. Moreover, we performed non-parametric tests with minimal assumptions about the data, and all statistical tests were corrected for multiple testing to account for the inflated occurrence of false positives. Finally, the biologists performed visual inspections, and the findings were compared to the literature to guarantee the meaningfulness of the results.

To address the above points, we added this paragraph in the Methods section:

Validation of the biological findings

In the study of the mode of action of antibodies, six donors for the CD19-TCB and seven donors for Teplizumab were analyzed, respectively. To avoid systematic bias in the analysis due to potential batch effects and emphasize the generalizability of the results, we included at least two independent experiments for each antibody: experiments III and IV for CD19-TCB and I, II, and III for Teplizumab. This allowed us to focus on consistent findings between the experiments. All statistical tests were corrected for multiple testing to account for the inflated occurrence of false positive results. Finally, visual inspections of the acquired images were done by expert immunologists, and the findings were validated in the literature.

We also addressed this in the discussion:

We generated the largest publicly available imaging flow cytometry data with over 2.8 million images using human primary immune cells from nine donors in four independent experiments that were treated with various therapeutic antibodies to study synapse formation. The large number of acquired images across multiple experiments provided sufficient statistical power to enable the study synapse formation, a potentially low-probability event, going beyond previous works that did, for example, not consider inter-experiments effects or inter- and intra- donor variability^{21,40}.

** As noted above regarding the limited number of annotator experts: please conduct a measurement for interobserver and intraobserver variability. Without such evidence, it is very difficult to appreciate a high level of fidelity for the dataset, and the subsequent controls for overfitting during model training/testing.*

Thank you for raising this point. We followed the reviewer's suggestion and performed an analysis on intra- and inter-rater comparability and provided all details above under comment #3.

** Explainable AI is not the same as having a list of human-readable features. Whether the authors agree or disagree with that statement, please provide supporting evidence and/or discussion.*

We agree and thank the reviewer for raising this important point. We will elaborate more on our approach since explainability for artificial intelligence is a complex topic heavily debated in the literature [29-30]. A common source for confusion is the interchangeable use of the terms explainable and interpretable and their association to either the machine learning model, the features (intrinsically learned or engineered), or both. With the term explainable Machine Learning, in our work, we refer to the explainability of the machine learning model. Generally,

one has to distinguish between explainability as an inherent characteristic of an algorithm or model or the explainability of so-called black box models, which can only be approximated by other methods [30]. Rudin discusses the shortcomings of post-hoc explainability methods applied to black-box artificial neural networks (ANNs), e.g., by using saliency maps in the imaging domain [60]. State-of-the-art ANNs, intrinsically learning classification boundaries, and suitable feature representations simultaneously often outperform classical machine learning methods combined with explicitly engineered features in terms of accuracy and performance. For instance, we applied Layer-wise Relevance Propagation (LRP) [61] to explain the ResNet34. As the figure demonstrates, LRP output only shows an important part of the image based on the network. However, this type of explainability needs to provide a clear biological insight into the problem at hand.

In our work, we found that using a classical machine learning algorithm, such as XGBoost, delivered performance on par with state-of-the-art ANNs (Figure 1c). XGBoost, an ensemble of decision trees, can provide a level of explainability based on variable importance [32]. Additionally, combining interpretable, human-readable features with the possibility to compute feature importances for the XGBoost model, as calculated using the GAIN or SHAP [32], we add a layer of interpretability for the user, allowing a deeper understanding of the model's decision-making. Such a design also provides a tool to understand the predictions of the model for the whole population or only a sample using post-hoc methods. This additional insight on interpretable features has the potential to generate new biological hypotheses or to derive streamlined, one or two-parameter decision criteria to be implemented in laboratory workflows.

To summarize these points, we now clearly define the type of explainability we refer to in this work in the Results section:

scifAI's explainable design follows the definition of Singh et al.²⁹ and Tjoa et al.³⁰, where machine learning models are explained by either in-model mechanisms, such as Gini-index, or by using post-model methods, such as saliency maps³¹ for deep learning models. Using interpretable features with a model whose internal mechanisms can be readily analyzed, scifAI natively provides in-model explainability and is fully compatible

with other libraries, such as SHAP³² or Captum³³, to provide further post-model explainability.

Additionally, we pick this up in the Discussion in more detail:

In this work, we followed the definition of Singh et al.²⁹ and Tjoa et al.³⁰ in the context of the explainability of machine learning and AI. We regard the biologically motivated features as interpretable as they are meaningful and can potentially hint at underlying biological mechanisms. We also considered our XGBoost model explainable as it natively provides a feature importance measure³². This design also enables a deeper understanding of the model's decision-making using additional methods, such as SHAP³². The insights from the interpretable features and explainable models have the potential to generate new biological hypotheses, which can lead to a better understanding of the underlying mechanisms at play. It is essential to mention that explainable models can offer a helpful intuition only if trained on interpretable features and will fail to provide a meaningful interpretation when trained on abstract features such as the bottleneck layer of an autoencoder. The combination of interpretable features and explainable machine learning enabled us to identify various relevant classes, such as immunological synapses, with state-of-the-art accuracy. It also allowed us to investigate the morphological profiles of the immunological synapse in an unbiased way and characterize the mode of action of antibodies in a biologically relevant context. This methodology is thus a substantial contribution to the field, which *has so far* primarily focused on performance over interpretability by using ResNet CNN architecture as the backbone^{26,41}.

[29] Singh, A.; Sengupta, S.; Lakshminarayanan, V. Explainable Deep Learning Models in Medical Image Analysis. *J. Imaging* 2020, 6, 52. <https://doi.org/10.3390/jimaging6060052>

[30] E. Tjoa and C. Guan, "A Survey on Explainable Artificial Intelligence (XAI): Toward Medical XAI," in *IEEE Transactions on Neural Networks and Learning Systems*, vol. 32, no. 11, pp. 4793-4813, Nov. 2021, doi: 10.1109/TNNLS.2020.3027314.

[32] Lundberg, S.M., Erion, G., Chen, H. et al. From local explanations to global understanding with explainable AI for trees. *Nat Mach Intell* 2, 56–67 (2020). <https://doi.org/10.1038/s42256-019-0138-9>

[60] Rudin, C. Stop explaining black box machine learning models for high stakes decisions and use interpretable models instead. *Nat Mach Intell* 1, 206–215 (2019). <https://doi.org/10.1038/s42256-019-0048-x>

[61] Wojciech Samek, Leila Arras, Ahmed Osman, Grégoire Montavon, Klaus-Robert Müller (2021): Explaining the Decisions of Convolutional and Recurrent Neural Networks, In: *Mathematical Aspects of Deep Learning*, pp. 1-33, Cambridge University Press, Cambridge, UK

In part two and part three of the "Results" session, "sciAI: An explainable AI python framework..." and "sciAI enables high-throughput profiling...", what stood out was: the authors implemented and benchmarked an XGBoost classifier that used pre-engineered handcrafted features, against other deep learning-based classifiers that contained difficult-to-explain representation embeddings. That work alone would not result in interpretability. For example:

We thank the reviewer for bringing up the important topic of explainability and interpretability and will answer point-by-point below to each comment.

- A single feature may have a clear name but does not have a clear meaning in biology: what does it mean when there are big or small cells in an immune reaction? Are big cells activated T cells or dividing cells or inactivated cells?

We completely agree with the reviewer in this regard. It is essential to underline that having an explainable classifier with meaningful features does not directly lead to interpretability. In our work, all the findings were closely investigated by experts in immunology to ensure their plausibility. In our view, scifAI can assist immensely in exploring biological processes or generating new hypotheses for follow-up analyses.

Furthermore, we try to answer the question of cell size in the context of cell type classification. In our case, we investigated synapse formation between EBV-transformed lymphoblastoid B-cells (B-LCL), an immortalized B cell line, and primary human memory T cells directly isolated from PBMCs without further activation applied in the assay. The sizes of B-LCL and T cells using IFC are distinct, potentially due to their different growth characteristics and lineages. Heterogeneity concerning cell size was only observed for B-LCL (bigger and smaller B-LCL cells) but not for T cells.

- A combination of meaningful features might result in better machine learning classification performance but might not have a clear biological reason. For example, big sizes and rough textures might be meaningful when they stand alone; but what does it mean the combination of both, is it a good or bad immune phenotype?

Answering this question depends on the biological context. Having multiple important features does not necessarily translate to a new finding. However, it should be used as inspiration for further investigation, and based on the biology, it should be decided if there is value in their combination.

The most useful discussion of the interpretability was presented at the end of part 3 of the "Results" session, where authors noted the most predictive features were co-localization, intensities, and textures of biomarkers (CD3, MHCII, P-CD3ζ etc.). However, please provide

further meaningful evidence of why the high/low of Kurtosis, Skewness, Contrast, and Correlation distance are good proxies of biological processes; e.g. are those high/low intensity-based features (and/or their combinatorial effects) equivalents of increase/decrease membrane protein expressions?

Considering that the context here is to predict the existing cell types in the images, we try to provide more insight about each feature:

- Statistics such as "skewness" examine the asymmetrical distribution of pixel intensities in the cells. If there is no clear expression, it will be zero. The more asymmetrical expression, the higher the skewness will be. For example, in Fig. 3d, if the skewness of P-CD3 ζ is low, it can indicate the lack of signal.
- Colocalization features like the "correlation distance of MHCII & CD3" indicate whether there is an overlap between CD3 and MHCII in the image, which can be translated to the proximity of the T cell and B-LCL in the image. For example, in Fig. 3d, the lowest value shows a complete overlap between the T cell and B-LCL, indicating the "B-LCL & T cell in one layer" class possibility. If the value grows, the overlap decreases, implying the classes with less overlap, such as "Synapse w/ signaling." Finally, suppose the value reaches its maximum. In that case, there is no overlap between a B-LCL and T-cell, hinting that the image might contain only one cell type.
- Texture features also hint at existing or absence of an expression in the expected fluorescent channel. For example, the low values of contrast of CD3 indicate the lack of a T cell in the image.

Based on these features, one can speculate that:

- The texture of CD3 and MHCII is used to detect the existence of T and B-LCL cells in the image.
- The colocalization of CD3 & MHCII is used to detect the different doublets types
- The intensity of P-CD3 ζ and colocalization of MHCII & P-CD3 ζ are used to detect whether there is a signaling T cell in the image (Fig. 1d).

In this specific use case, these features only help "detect" the cell types in the images and do not provide further insights about the quality of the synapses. Therefore, we dedicated a major part of the text to the mode of action of therapeutic antibodies (Fig. 2&3)

** The authors first noted "channels BF, MHCII and P-CD3 ζ sufficed to reach an F1-macro...", but later in the part "Characterizing the impact of therapeutic antibodies on synapse formation" indicated that "[BF] intensity is difficult to interpret and its morphological characteristics can be captured also by other fluorescent channels". This is contradictory:*

We thank the reviewer for pointing out the ambiguity in the text. We have addressed them point-by-point below each comment.

- How could an explainable AI framework have room for the exclusion of difficult-to-interpret features?

We thank the reviewer for bringing up this point. Based on the work of Tolosi et al. [1] and Strobl et al. [2], it is desirable to have the possibility of exclusion of features before any analysis. This helps avoid skewing the results based on noisy or meaningless features.

[1] Laura Toloși, Thomas Lengauer, Classification with correlated features: unreliability of feature ranking and solutions, *Bioinformatics*, Volume 27, Issue 14, July 2011, Pages 1986–1994, <https://doi.org/10.1093/bioinformatics/btr300>

[2] Strobl, C., Boulesteix, AL., Kneib, T. et al. Conditional variable importance for random forests. *BMC Bioinformatics* 9, 307 (2008). <https://doi.org/10.1186/1471-2105-9-307>

- Secondly, if BF could be easily replaceable, why would the author praise its usefulness in the first instance?

We would like to emphasize that these are two different types of analyses. The first analysis aims to find the minimal staining panel to train a machine learning model that can predict the images' cell types. Here, BF is used as it is stain-free and is captured for free by the imaging flow cytometer. We added a paragraph to elaborate on this:

Next, we investigated the effects of fluorescent channels on the classification performance to determine which antibodies are essential for the detection of synapses and which ones can be freely exchanged depending on the biological context. Considering that BF is a stain-free channel provided for free by IFC, we kept the BF channel fixed and added all possible combinations of the fluorescent channels to train the model. We found that the combination of the channels BF, MHCII, and P-CD3ζ sufficed to reach an F1-macro of 0.91 ± 0.01 (Supplementary Fig. 5b), similar to using all the channels (0.92 ± 0.01).

In the second analysis, we aim to understand the mode of action of antibodies. Considering that fluorescent channels include more targeted information in terms of mode of action, we mainly focused on them. We agree with the reviewer that this was unclear from the original text. Therefore, we modified the manuscript to explain this point more clearly:

~~We avoided using the BF channel as its intensity is difficult to interpret and its morphological characteristics can be captured also by other fluorescent channels.~~ Considering that we were interested in the mode of action of antibodies, we focused on fluorescent channels providing targeted information on components of the cell expected to change during synapse formation morphologically.

** The results of CD19-TCB and Teplizumab on synapse formation, with its associated Figure 3 and discussion were great. Appreciate the detailed presentation.*

Thank you!

** Exploring the use of image features in predicting cytokine production is ambitious, and it is unconvincing when the number of samples is low, as noted by the authors themselves. While the attempts are appreciated, please further prove the alignment of the "standard deviation of MHCII (95th perc.) [intensity]" and "eccentricity of F-actin (95th perc.) [morphology]" with the biological process of cytokine production.*

As correctly pointed out by the reviewer, we found intensity of MHCII and morphology of F-actin as the most prominent features for TCBs in predicting cytokine production. However, it is not possible to reach a finding generalizable to a larger cohort due to the limited number of donors. Nevertheless, these two features could indeed be biologically linked to processes that are involved in cytokine production. Our data shows that TCBs drive immune synapse formation, however, interaction between MHCII and TCR can still occur and potentially contribute to amplify TCR signaling and T cell polarization [1]. Hence, MHCII accumulation can still be meaningful. The eccentricity of F-Actin resembles the closeness of the T cell and B-LCL towards each other and it is known that the rearrangement of the actin cytoskeleton is essential for the development of mature immune synapses [2]. Both features can be used as inspiration for further investigation and formulation of new hypotheses on the mode of action of TCBs. Nonetheless, additional experiments are necessary to validate them.

[1] Sophie Duchez, Magda Rodrigues, Florie Bertrand, Salvatore Valitutti; Reciprocal Polarization of T and B Cells at the Immunological Synapse. *J Immunol* 1 November 2011; 187 (9): 4571–4580. <https://doi.org/10.4049/jimmunol.1100600>

[2] Burkhardt JK, Carrizosa E, Shaffer MH. The actin cytoskeleton in T cell activation. *Annu Rev Immunol.* 2008;26:233-59. doi: 10.1146/annurev.immunol.26.021607.090347. PMID: 18304005.

Minors:

** Please spell out the first instances of abbreviations, or make an effort for a glossary table, such as "MHC", "EBV", "FACS", "PBMCs" etc.*

We added all abbreviations in the main text and a glossary in the supplementary Table 3.

** There was an inconsistency in how the manuscript writers indicate cellular conjugates: sometimes "T:APC", other times "T-B"*

Thank you for noticing the inconsistencies in using the terms T:APC and T-B cell conjugates. We changed “T cell-APC interactions” to “**T-B cell interactions**” in the Discussion because we address the function of Teplizumab in our T-B cell system, and it’s good to be more precise in the description. Nevertheless, we stuck to “T-APC conjugates” in the introduction because we cited publications that show conjugates with B cells and macrophages, which are both APCs.

** Figure preparations were great, but figure captions were not, for example, Figure 2 captions are inadequately described. "a, c Schematic representation of the mode of action ...", please describe those modes of action. Similar expansions to supplementary figures' captions are also needed.*

We worked on the figure captions and explained them in more detail, especially focusing on explaining the mode of action in Figure 2a and 2c.

Figure 2. CD19-TCB and Teplizumab show significant changes in the frequencies of synapses

a, c Schematic representation of the mode of action of CD19-TCB and Teplizumab. CD19-TCB binds with one arm to the T cell receptor CD3 and with two arms to the B cell co-receptor CD19 thereby bringing T and B cells into proximity and activating T cells. Teplizumab binds with two arms to CD3 and has been described to inhibit T cell activation. To exert their full suppressive function, the T cells needed to be first stimulated via the superantigen SEA

** In the "Introduction" session, second paragraph, last sentence "IFC has recently been successfully ... the immunological synapse of ...T:APC cell, however, none of these investigated ... the immunological synapse in the context of T cell function". This important sentence is confusing, what is the actual difference between the first and the second mentioning of "immunological synapse".*

Thank you for mentioning this important sentence. We changed the wording “T cell function” to “**T cell effector function**” in the introduction because it better describes the function of T cells after the formation of the immunological synapse and their activation. So far, no other study has looked deeper into T cell effector functions like cytokine production. In our work, we cover early events of T cell activation, like TCR signaling induced by the formation of the immunological synapse and how it is translated into T cell effector functions like cytokine production after 24h.

Reviewer #3 (Remarks to the Author):

The work establishes an open-access software scifAI based on imaging flow cytometry and explainable machine learning to predict the functionality of therapeutic antibodies in vitro. It also includes 2.8 million IFC cell images of human immunological synapse and makes these images available to the public. This work uses interpretable features of the immunological synapse to predict the effectiveness of therapeutic antibodies on T cell cytokine production to allow prediction of the functional outcome of an unseen antibody. There are many positive points, making the paper a worthy addition to the scientific literature. However, several major issues need to be addressed before acceptance of the work for publication:

1. The authors did not discuss whether the cell images and the explainable AI algorithm are dependent on the image modalities or specific brand/model of IFC. Can the image data be used to train cell images from different imaging equipment such as a confocal microscope or different models of imaging flow cytometer? Can these images in the database be used to train cell images with different magnification factor, wavelengths, or focal depth? Basically, the authors need to define and specify any constraints of the usability of their data.

We thank the reviewer for the positive feedback and issues raised. Concerning the robustness of our model to other data, we followed the reviewer's suggestion and elaborated on usability constraints in the revised Discussion of our manuscript, where we now write:

While we demonstrate good generalizability of our model, we do not expect that the model trained on our specific IFC data will be predictive without additional training on different datasets obtained with other IFC machines out of the box due to the inherent domain shifts in resolution, magnification, light wavelengths, or focal depth. However, exploiting transfer learning, self-supervised pre-training⁴¹, and domain adaptation and generalization techniques⁴², this dataset will be a valuable resource for future applications, such as transfer to other fluorescent imaging modalities, for example, confocal microscopy, where data can be scarce. Training deep generative models to understand the mode of action of therapeutic antibodies⁴³ is another exciting avenue for future research.

Above that, we would like to note that we have already used our scifAI framework for other datasets and added the following sentence to the Discussion:

Notably, the scifAI framework can be easily trained with other fluorescent imaging data. As proof of concept, we provide three examples within the scifAI code repository on how to analyze IFC datasets from Jurkat cells³⁴, white blood cells²², and apoptotic cells²⁴ using scifAI.

2. The authors mention that they throw away some images for different reasons (out of focus, doublets, blur, etc.). It would be helpful for readers to understand what images are "acceptable" and what should be "rejected." Is there a set of quantitative criteria to determine the qualified images for the machine learning model?

We thank the reviewer for this remark. Accordingly, we revised the Method's 'Data cleaning pipeline' section, where we now more clearly and concisely lay out our feature-based rules:

For each donor, we used the trained XGBoost classifier to predict the class of every image. We excluded low-quality and outlier images based on the following data cleaning protocol, where we first describe the types of cells we filtered out, followed by the concrete feature-based rule:

1. Dead cells (using Live/Dead staining): 'mean Live/Dead intensity' \geq 'mean Live/Dead intensity (90th perc.)'
2. Out-of-focused images: 'Gradient RMS BF' $>$ 'Gradient RMS BF (2nd perc.)' & 'Gradient RMS BF' $<$ 'Gradient RMS BF (90th perc.)'
3. High entropy images, based on the XGBoost predictions: entropy $>$ 1.0. The entropy was calculated using the SciPy package. This step is done to omit images that the classifier is the most uncertain in terms of prediction.
4. Images predicted as 'B-LCL' with low MHCII: 'mean intensity of MHCII' $<$ 'mean intensity of MHCII (5th perc.)'. This step guarantees that the images predicted as 'B-LCL' contain a minimum of MHCII intensity.
5. 'B-LCL' with 'area of MHCII' $<$ 'area of MHCII (10th perc.)'. This step guarantees that the images predicted 'B-LCL' contain a cell with appropriate size.
6. 'T cell' with 'mean intensity of CD3' $<$ 'mean intensity of CD3 (1st perc.)'. This step guarantees that images predicted as 'T cell' contain a minimum CD3 intensity.
7. 'B-LCL and T cell in one layer' with small 'B-LCL's: 'B-LCL and T cell in one layer' with 'area of MHCII' $<$ 'area of MHCII (20th perc.)'. This step is performed to omit missclassified 'B-LCL and T cell in one layer' with small 'B-LCL's.
8. Outlier detected via isolation forest ⁵⁷: We used `n_estimators=100`, `max_samples='auto'`, `contamination='auto'`, and `max_features=20` as the main parameters. For reducing the run time, we only used top 30 features based on feature importance from the XGBoost training.
9. Outlier detected via DBSCAN ⁵⁸: First we transformed all images to 2D dimensional space using Uniform Manifold Approximation and Projection (UMAP). The features were standardized using the mean and std of each feature. For reducing the run time, we only used top 30 features based on feature importance from the XGBoost training. Then a DBSCAN algorithm was run with `eps=0.09` and `min_samples=5`. The resulted clustered were filtered out if $(\#images\ in\ cluster)/(\#total\ images) < 0.0001$.

All these steps are based on scikit-learn implementations. All parameters were set using the default value of scikit-learn unless stated otherwise. 20 random examples of filtered-out images are shown in Supplementary Fig. 6.

Additionally, we refer to the data cleaning explicitly in the main text, where we now write:

“...A data cleaning pipeline was also implemented to filter out unwanted images such as experimental artifacts (Methods and Supplementary Fig. 6).”

Finally, we added Supplementary Figure 6 with examples of excluded images:

Supplementary Figure 6. Filtered-out images

20 randomly selected examples that were excluded from the analysis based on the data cleaning pipeline described in Methods. The examples show dead cells (with signal in the “Live/Dead” channel), out-of-focus images and images with irregular shapes in the brightfield channel, or missing fluorescent expression (scale bar = 2.4 μ m)

3. The paper put great emphasis on "explainability" of the AI results. Explainable AI is obviously the key component of the paper. However, it is not clear to the reviewer how the "explainability" is measured and verified. The paper shows agreement between human recognizable image features (e.g. co-localization) and AI results, but the correlation between the two does not necessarily indicate that this is how AI uses to produce its results. There is no analysis such as CAM to better assure or elucidate AI explainability.

We thank the reviewer for this critical remark. From our perspective, explainability in the present study comes from using interpretable features with a model whose internal mechanisms can be readily analyzed [30]. With respect to methods that highlight pixel-wise importance, like CAM, mentioned by the reviewer, we would like to mention that these methods are not designed for a feature-based approach, like the one used in scifAI. Moreover, their value model for explainability is limited and has been recently critically discussed (e.g. [60]). Following the reviewers' remarks, we applied Layer-wise Relevance Propagation (LRP) [61], a state-of-the-art interpretability method to the ResNet34 model. LRP output highlights cells in the image but fails to provide more biological insight into the problem at hand.

To incorporate the reviewer's comment regarding the lack of clarity in the context of explainability, we elaborate more on the definition of explainability in the manuscript. First, it is addressed in the design paradigm of scifAI in the Results where we now write:

scifAI's explainable design follows the definition of Singh et al. ²⁹ and Tjoa et al. ³⁰, where machine learning models are explained by either in-model mechanisms, such as Gini-index, or by using post-model methods, such as saliency maps ³¹ for deep learning models. Using interpretable features with a model whose internal mechanisms can be readily analyzed, scifAI natively provides in-model explainability and is fully compatible with other libraries, such as SHAP ³² or Captum ³³, to provide further post-model explainability.

Additionally, we pick this up in the discussion in more detail, where we now write:

In this work, we followed the definition of Singh et al.²⁹ and Tjoa et al.³⁰ in the context of the explainability of machine learning and AI. We regard the biologically motivated features as interpretable as they are meaningful and can potentially hint at underlying biological mechanisms. We also considered our XGBoost model explainable as it natively provides a feature importance measure³². This design also enables a deeper understanding of the model's decision-making using additional methods, such as SHAP³². The insights from the interpretable features and explainable models have the potential to generate new biological hypotheses, which can lead to a better understanding of the underlying mechanisms at play. It is essential to mention that explainable models can offer a helpful intuition only if trained on interpretable features and will fail to provide a meaningful interpretation when trained on abstract features such as the bottleneck layer of an autoencoder. The combination of interpretable features and explainable machine learning enabled us to identify various relevant classes, such as immunological synapses, with state-of-the-art accuracy. It also allowed us to investigate the morphological profiles of the immunological synapse in an unbiased way and characterize the mode of action of antibodies in a biologically relevant context. This methodology is thus a substantial contribution to the field, which *has so far* primarily focused on performance over interpretability by using ResNet CNN architecture as the backbone^{26,41}.

[29] Singh, A.; Sengupta, S.; Lakshminarayanan, V. Explainable Deep Learning Models in Medical Image Analysis. *J. Imaging* 2020, 6, 52. <https://doi.org/10.3390/jimaging6060052>

[30] E. Tjoa and C. Guan, "A Survey on Explainable Artificial Intelligence (XAI): Toward Medical XAI," in *IEEE Transactions on Neural Networks and Learning Systems*, vol. 32, no. 11, pp. 4793-4813, Nov. 2021, doi: 10.1109/TNNLS.2020.3027314.

[32] Lundberg, S.M., Erion, G., Chen, H. et al. From local explanations to global understanding with explainable AI for trees. *Nat Mach Intell* 2, 56–67 (2020). <https://doi.org/10.1038/s42256-019-0138-9>

[60] Rudin, C. Stop explaining black box machine learning models for high stakes decisions and use interpretable models instead. *Nat Mach Intell* 1, 206–215 (2019). <https://doi.org/10.1038/s42256-019-0048-x>

[61] Wojciech Samek, Leila Arras, Ahmed Osman, Grégoire Montavon, Klaus-Robert Müller (2021): Explaining the Decisions of Convolutional and Recurrent Neural Networks, In: *Mathematical Aspects of Deep Learning*, pp. 1-33, Cambridge University Press, Cambridge, UK

Reviewer #4 (Remarks to the Author):

The authors generate an imaging flow cytometry dataset of B-T cell conjugates using CD4 T cells, B lymphoblastoid cells and superantigen and develop a machine learning-based method for the classification of the images to detect the formation of immunological synapses and extract biological information, in order to evaluate the function of different therapeutic agents. The work is interesting and novel. However, it is not clear how generalizable it is for the work of other users, with different types of samples and/or biological questions. Also, it might be challenging to use for biologists without programming knowledge, therefore limiting the potential impact of the method.

We would like to thank the reviewer for this important comment on the generalizability of our model, which has also been raised by reviewer 3. Following both reviewers' remarks, we elaborated on usability constraints in the revised Discussion of our manuscript, where we now write:

While we demonstrate good generalizability of our model, we do not expect that the model trained on our specific IFC data will be predictive without additional training on different datasets obtained with other IFC machines out of the box due to the inherent domain shifts in resolution, magnification, light wavelengths, or focal depth. However, exploiting transfer learning, self-supervised pre-training⁴¹, and domain adaptation and generalization techniques⁴², this dataset will be a valuable resource for future applications, such as transfer to other fluorescent imaging modalities, for example, confocal microscopy, where data can be scarce. Training deep generative models to understand the mode of action of therapeutic antibodies⁴³ is another exciting avenue for future research.

We would like to note though, that we have already used our scifAI framework for other datasets with different biological questions, detailed in our GitHub repository, and added the following sentence to the Discussion:

Notably, the scifAI framework can be easily trained with other fluorescent imaging data. As proof of concept, we provide three examples within the scifAI code repository on how to analyze IFC datasets from Jurkat cells³⁴, white blood cells²², and apoptotic cells²⁴ using scifAI.

Moreover, we have added the following paragraph to the Methods of our revised manuscript to address the point of code usability:

Reproducing the results

For reproducing the results or running exemplary code provided as part of the software package, it is necessary to download the dataset and install scifAI. Extensive documentation

has been provided that will allow users with basic programming knowledge to follow four application examples and, with minor modifications, adapt the code to their needs. To tackle more advanced use cases, such as defining a new feature, or applying scifAI to a completely new data set, more programming experience is required.

Major concerns include:

- Authors use a stratified sampling scheme which guarantees that the training set will be representative of all donors used for testing. It seems that training was not limited to the initial SEA driven synapse datasets, but was continuous with mention of keeping some types of data in the “training set”. Does the model generalize for donors that are not part of the training set? Authors should clarify that the model is transferable between different donors by using different donors for training than for testing.

- Similarly, the problem dataset and the training dataset are part of the same bigger set of images and this could cause overfitting of the model. How robust is the model for datasets more different to the training set?

We appreciate this comment, and we completely agree with the reviewer to discuss model robustness as clearly as possible. We performed a new experiment with leave-one-donor-out cross-validation to confirm inter-donor generalizability. This is reflected in the Results section of the revised manuscript, where we now write:

To confirm the validity of the results, a leave-one-donor-out cross-validation using the interpretable features and XGBoost was performed. The cross-validation yielded F1-macro values of 0.88 ± 0.04 , demonstrating good generalizability across donors (Supplementary Fig. 4d).

We also added a new supplementary figure to support the text:

Supplementary Figure 4d Leave-one-donor-out cross validation confirms that the result in c is robust (compared to stratified \pm SEA-based train and test set)

Throughout our work, we have validated our results rigorously. To emphasize this more, we have now clearly defined our three validation sets as \pm SEA-based, CD19-TCB-based, and Teplizumab-based. We added a paragraph in the Methods section for more clarity on this important point:

Classification performance validation and generalizability

We validated the prediction performance of classification models across multiple datasets. To offer a comprehensive view of the generalizability of our results, we present a complete list of all the steps taken. The first validation was performed on the \pm SEA-based test set (30% hold-out set) to confirm inter-experiment comparability (Supplementary Fig. 4c). The second validation was performed using leave-one-donor-out cross-validation to confirm inter-donor generalizability (Supplementary Fig. 4d). Next, the model was validated on two separate test sets with Teplizumab-based and CD19-TCB-based perturbations to validate inter-experiment and inter-stimulation generalizability (Supplementary Fig. 7). Overall, we demonstrated that the XGBoost model trained on \pm SEA perturbations could indeed generalize to new perturbations with previously unseen antibodies and inter-donor variability.

- How robust is it to day to day differences in instrument settings and have data subjected to analysis been collected on different machines. For example, does the model still work if using different image acquisition parameters (for example different laser power) or different types of samples, for example different subsets of B and T cells (like naive T cells (smaller) or blasted T cells (bigger))?

We thank the reviewer for this important point. For a new experiment with different parameters or other biological questions, scifAI is still applicable. However, there is no guarantee that the already trained model will perform well on the new data. We addressed this limitation, common to many if not all current machine learning models, in the revised Discussion:

While we demonstrate good generalizability of our model, we do not expect that the model trained on our specific IFC data will be predictive without additional training on different datasets obtained with other IFC machines out of the box due to the inherent domain shifts in resolution, magnification, light wavelengths, or focal depth.

To emphasize the applicability of scifAI to other datasets, we have provided examples in our documentation (<https://github.com/marrlab/scifAI/tree/main/docs>) and added the following paragraph to the Discussion:

Notably, the scifAI framework can be easily trained with other fluorescent imaging data. As proof of concept, we provide three examples within the scifAI code repository on how to analyze IFC datasets from Jurkat cells ³⁴, white blood cells ²², and apoptotic cells ²⁴ using scifAI.

- It is impressive that the AI was able to correctly score both activation and inhibition of synapse formation and function. To better understand the outputs, is it correct that the scoring is all 0 or 1, no or yes, and the continuous data in, for example, Fig 3b, d, f, etc is an average of the binary results. Did the training consist of making these binary judgements or of indicating where to make quantitative measurements? As a biologist, I would like to take advantage of such information and my impression is that the more automated feature selection was performing some quantitative measurements with %tiled measurements, but were these measurement (such as Fig 3 a, h) only made after the classification using the human training data into the different major categories? I would appreciate if this could be clarified.

We thank the reviewer for pointing out this lack of clarity. We would like to emphasize that there was no direct training involved in Figure 3. To elaborate, we first used the model from the previous part (\pm SEA-based trained) and predicted the class of every image for CD19-TCB, Ctrl-TCB, Teplizumab, and Isotype. Then we only selected the images identified as “synapse w/ signaling.” Next, we compared the features of antibodies and their corresponding controls. The comparison is made using the Mann–Whitney U test. In order to make this clear in the manuscript, we have elaborated on this important point further in the Methods of our revised manuscript:

Feature difference analysis

We analyzed the effect of perturbation with CD19-TCB, and Teplizumab on signaling synapses using the previously trained XGBoost model from the \pm SEA training set and then used statistical tests to investigate the mode of action of these therapeutic antibodies. First, we selected the images predicted as ‘synapse w/ signaling.’ We only focused on fluorescent channels as they contain targeted information on components of the cell expected to morphologically change during synapse formation. This procedure yielded 210 features for comparison for TCB based on F-actin, MHCII, CD3, and P-CD3 ζ . For Teplizumab, CD3 was not available for the analysis because of the usage of CD4 in recording images for Teplizumab instead of CD3. Thus we analyzed 132 features extracted from F-actin, MHCII, and P-CD3 ζ . After the feature selection, we compared the features using the Mann-Whitney U test for each condition and its control. To understand the direction of change, we used the difference in the median of features for each condition and its control. To account for multiple testing, we used the Benjamini-Hochberg procedure with $\alpha=0.05$. As the conditions were independent, we corrected the p-values for each condition and its control separately. If the test was not significant, we assigned that feature 0 (gray in Fig. 3a,h and 0 in Supplementary Tables

1-2). If it was significant and the median value of the feature for the antibody was greater than the median of the feature of the control antibody, we assigned +1 (red in Fig. 3a,h and +1 in Supplementary Tables 1-2). On the contrary, if the test was significant, but the median value of the feature for the antibody was smaller than the median value of the feature of the control, we assigned -1 (blue in Fig. 3a,h and -1 in Supplementary Tables 1-2).

Minor:

- Fig 4a: legend mentions colors that are not present in the image.

- Suppl Fig 5: legend includes more panels than shown in the figure.

We thank the reviewer for the attention to detail and apologize for the inconsistencies. In the revised version of our manuscript, we made adjustments accordingly.

Reviewers' Comments:

Reviewer #1:

Remarks to the Author:

After the last round of review, the author made fruitful revisions to the manuscript. However, I do not think this work is worth being published in Nature Communications in its current form:

1. The author should elaborate more details about the relationship between the biologically motivated features and the biological mechanisms, and the relationship between the XGBoost model and the feature importance measure.
2. In the paragraph "scifAI is an end-to-end data acquisition...and the functionality of newly generated antibody candidates." in the Discussion. With some examples only, the author foresees that scifAI can even help to identify responders among patient populations and predict their clinical outcomes. However, this prediction lacks early data as evidence to support this view.
3. The dataset comes from nine donors, whether it can represent the whole field of therapeutic antibodies.
4. The explainable machine learning utilizes morphology, intensity, co-localization, texture, and synaptic features, why choose these features?
5. The scifAI framework for the efficient and explainable analysis of high-throughput imaging data, and high-throughput is a highlight in this paper. However, the paper does not show the quantitative analysis data to support the point.
6. The dataset contains over 2.8 million images, but for the training process, this paper only uses 5221 images instead of the whole dataset.
7. In 'A subset of annotated data and available IFC channels suffices for a high classification performance', the paper demonstrates the relationship between the annotated ratio and the performance, but the point only tests on the small 1500 images and lacks theory. Does it is reasonable for all datasets?

Reviewer #2:

Remarks to the Author:

I appreciate the authors' efforts to revise the manuscripts, conducted additional experiments, provided additional evidence and data, and significantly improved the quality of the research. The rebuttal to reviewers was also well-prepared, and fairly addressed my previous concerns. However, in my opinion, the revised article still could not yet meet the expectations of this journal tier:

- 1) In the current form of the article, two separate teams of scientific contributions could be observed: (A) one from a team of data analysts and machine learning experts, who contributed computational methodologies, model developments/validations and discussions, and analytics codes; and (B) one from a team of data generators, clinical immunologist(s) and annotators. There is however a mismatch between what (A) wishes to achieve and what (B) could provide. An example is the experimental setup for intra- and inter-rater variability testing: 100 labeled samples were selected to challenge 03 raters (one with an immunology background, and two data analysts). Readers will not be able to appreciate the fidelity of the results based on such a small test set, with the inclusion of only one biologist.
- 2) The added text for explainable AI was fairly helpful, including Singh et al. (29) and Tjoa et al. (30). On the positive side, the authors showed attempts to deliver: (A) the list of human-readable feature names, (B) a model that could natively provide feature importance measure, and (C) biological meaning of the features. However, it is not yet sufficient to claim this explainability is

novel in the field and better than state-of-the-art. More direct comparison, benchmark, or reference to truly state-of-the-art studies in 2019-2023 is needed for such a claim. Secondly, most of these explainable imagery features are presented through the lens of Amnis ImageStreamX Mark II Imaging Flow cytometer, an instrument that had its first debut around ~2010s (Imaging flow cytometry technology itself is even older).

3) There was more than one reviewer who noticed the generalizability issue of the model. The authors professionally conducted several validation settings for the XGBoost and ResNet. However, it is questionable why only XGBoost could be this good for XAI, in general, and in this research in particular.

Reviewer #3:

Remarks to the Author:

The authors provide satisfying responses and revisions of the paper in response to the reviewers' comments.

Reviewer #4:

Remarks to the Author:

The authors have provided helpful additional explanation of the meaning of explainable AI vs interpretation and other points of common concern to the reviewers. The discussion of how generalisable this dataset could be perhaps gives up on this too easily. As far as I'm aware there is really one instrument that does imaging flow cytometry. The staining reagents that are being used are also very standard commercial reagents. Could the author envision a pipeline to calibrate imaging conditions to guide a user to adopt conditions that would allow a human peripheral blood T cell to B-LCL conjugate system to brought into range with the parameters to the trained model as downloaded from GitHub? This might be similar to a bead calibration to generate consistent results in flow cytometry, but a step beyond this to bring all staining conditions and machine setting into a range where the AI could correctly segment the cells and synapse. Perhaps this could be pursued with Amnis as this reverse training could also be helpful to standardise machines and generally support reproducibility in a community. I don't expect you to address here, but might be worth considering this in the future. Otherwise I have no further concerns.

REVIEWER COMMENTS

Reviewer #1 (Remarks to the Author):

After the last round of review, the author made fruitful revisions to the manuscript. However, I do not think this work is worth being published in Nature Communications in its current form:

1. The author should elaborate more details about the relationship between the biologically motivated features and the biological mechanisms and the relationship between the XGBoost model and the feature importance measure.

We thank the reviewer for this suggestion. To address this concern, we added a paragraph with a detailed list of features as well as references to shed more light on the biological interpretation:

In order to characterize the immunological synapse in an unbiased fashion, we first designed and computed a series of biologically motivated, interpretable features using the scifAI framework. These features were based on morphology, intensity, co-localization, texture, and synaptic features extracted from the 5-panel stained images and their corresponding masks (Methods and Supplementary Fig. 3a-c). Morphology features included shape-related characteristics such as area, perimeter, and roundness of cells. Intensity and texture features were implemented to quantify the existence, distribution, and regularity of intensities within cells¹. Co-localization features were designed to capture similarities and differences in intensities among different channels². Finally, synaptic features were implemented to give insights into the distribution of intensities within the synapse region compared to the rest of the cell^{3,4}. Synaptic features were implemented based on the signal intensity ratio of each fluorescent channel in the synaptic area to the whole cell. These features allowed for comparing cell states within and among different populations.

As suggested, we rewrote the explainability paragraph, including more details about the intrinsic mechanism used. In addition, we added a new state-of-the-art explainability method, SHAP, as a sanity check, as well as the corresponding figures and the reference to a recent study that shows SHAP is the most used method in related topics:

Next, we focused on the explainability of the pipeline and explored which underlying features drive the class prediction, ranking features by their respective feature importance. The feature importance was based on the in-model mechanism called gain⁵, which signifies the relative contribution of a corresponding feature to the classification (Methods). The most predictive

features were based on the intensity of CD3, co-localization of MHCII & P-CD3 ζ , co-localization of MHCII & CD3, the cell morphology in BF, and the intensity of the MHCII (Fig. 1d). The intensities of CD3 and MHCII imply the presence of a T cell or B-LCL in the image. The co-localization of MHCII and P-CD3 ζ measures how many overlapping pixels the two proteins share. For example, `synapse w/ signaling` has a lower overlap between the T cell and the B-LCL than the `T cell & B-LCL in one layer,` and thus a lower co-localization of MHCII and P-CD3 ζ . The same logic can be applied to the co-localization of MHCII and CD3. The equivalent diameter of BF hints at the size of the cells or the existence of multiple cells. Based on the features and the definition of classes, one could speculate that the classifier uses (i) the intensity of CD3 and MHCII to detect the existence of T and B-LCL cells in the image, (ii) the co-localization of MHCII & P-CD3 ζ as well as MHCII & CD3 to detect the different doublets types and the existence of signaling (iii) the cell morphology in BF to assist detecting the cell type (Fig. 1d). To validate the feature importance with a widely used, post-model explainability approach, SHAP ⁶ values were calculated (Methods). SHAP is a game theory-based approach that has recently been shown to be the most used post-model explainability method in related topics ⁷. The top-5 SHAP values (Supplementary Fig. 5c) were similar to the previously shown feature importances based on model-intrinsic gain, providing additional confidence in the stability of the feature importance estimation (Fig. 1d).

2. In the paragraph "scifAI is an end-to-end data acquisition...and the functionality of newly generated antibody candidates." in the Discussion. With some examples only, the author foresees that scifAI can even help to identify responders among patient populations and predict their clinical outcomes. However, this prediction lacks early data as evidence to support this view.

We appreciate the raised concerns and deleted the respective sentence.

3. The dataset comes from nine donors, whether it can represent the whole field of therapeutic antibodies.

Clearly, the response from nine donors is not representative of the whole spectrum of therapeutic antibody responses. While we agree with the reviewer on this limitation, we would like to highlight two points:

- Compared to previous works on profiling immune synapses using machine learning and imaging data, we provide the largest dataset regarding the number of samples and the highest number of donors. For example, *German et al. (2021)*⁸ used 9 donors with fewer samples and lower

throughput, *Xiong et al. (2018)*⁹ used 5 donors, and *Naghizadeh et al. (2022)*¹⁰ used only 2 donors.

- The main novelty of our work is to provide a framework to develop an end-to-end pipeline for studying immunological synapses and their functionality. This work is thus the first coherent functional study using large image data and the subsequent analytics to visualize, understand, and study synapse formation and its link to predictive features.

4. The explainable machine learning utilizes morphology, intensity, co-localization, texture, and synaptic features. Why choose these features?

We aimed to capture a variety of aspects relevant to synapse formation. A detailed answer to the relationship between the features and biological mechanisms is provided in response to issue #1.

5. The scifAI framework for the efficient and explainable analysis of high-throughput imaging data and high-throughput is a highlight in this paper. However, the paper does not show the quantitative analysis data to support the point.

Throughout this work, we have extensively used and analyzed high-throughput data quantitatively and illustrated the results in Figures 2, 3, and 4 in detail. Nonetheless, we seized the opportunity to add a paragraph in the Results section, providing more motivation about developing scifAI, including relevant references.

High-throughput imaging flow cytometry enables systematic profiling of millions of cells, thus providing a valuable resource for gaining biological insights¹¹. Full manual annotation of such large data sets is prohibitive as expert time is scarce and expensive^{12,13}. Hand-crafted gating strategies, commonly used in IFC applications¹⁴, are hard to reproduce, often subjective and biased, and time-consuming for extensive experiments^{3,15}. Additionally, it has been shown that they can be suboptimal in prediction performance^{16,17}. In order to overcome these limitations, we developed the single-cell imaging flow cytometry AI (scifAI) framework for the unbiased analysis of high-dimensional high-throughput IFC data.

We also added the following sentence in the Discussion section to address the reviewer's comment regarding the efficiency of the pipeline.

On a 24-CPU machine, scifAI enables feature extraction and class-label prediction of approximately 250,000 images per hour. This is, by orders of magnitude, more efficient than manual annotation, as reported by our annotators, with a rate of approximately 100 images per hour.

6. The dataset contains over 2.8 million images, but for the training process, this paper only uses 5221 images instead of the whole dataset.

In applied supervised machine learning, annotations for biomedical images are often scarce as they crucially depend on the availability of trained experts, whose time is expensive and limited^{12,13}. Therefore, it is common practice to train models on small subsets of annotated data and subsequently predict on larger, unannotated data^{13,16,18}. Still, the reviewer's remark is interesting, and to exploit a large amount of unlabelled data, we now additionally used Barlow Twins¹⁹, a self-supervised learning algorithm for "pre-training" the ResNet18 and ResNet34 based on all unlabeled data. We added the following section in the Results:

We also trained a number of convolutional neural network (CNN) architectures, such as Resnet18, ResNet34, and DeepFlow, which had previously been shown to be successful in classification tasks on imaging flow cytometry data^{16,20,21}. For Deepflow, a random initialization was used. A self-supervised method called Barlow Twins¹⁹ was used to pre-train ResNet18 and ResNet34. Here, the unlabeled part of the \pm SEA-based was utilized based on Barlow Twins self-supervision task, and then the weights were transferred for the supervised training. This method has been shown to improve the performance of the CNN models in classification tasks¹⁹.

Moreover, based on the previously presented results, we trained an autoencoder on the unlabeled set. Therefore, the unlabeled data has been used in the training process. However, it did not provide a considerable performance boost compared to supervised learning only on the subset of labeled data.

7. In 'A subset of annotated data and available IFC channels suffices for a high classification performance,' the paper demonstrates the relationship between the annotated ratio and the performance, but the point only tests on the small 1500 images and lacks theory. Does it is reasonable for all datasets?

We would like to emphasize that while our paper is applied machine learning work, the training and testing of our model with the available data is in line with the current state of the art in machine learning research and well based on the latest theoretical findings. Our test set contains 33% of the annotated dataset based on multiple donors and experiments, constituting a reasonable random sample of the data representative of the full data set. While it is obviously beneficial to have large quantities of annotated data available for any machine learning problem, in many real-world applications, there exist strong limitations due to the scarcity of experts' time. It has also been shown that annotating the whole dataset is not always necessary, and such "incomplete supervision" has been well-studied by the community²². Moreover, it has been shown in other applications that by annotating a

subset of the data, it is possible to achieve a reasonably high performance compared to annotating the whole dataset ^{12,13}.

Reviewer #2 (Remarks to the Author):

I appreciate the authors' efforts to revise the manuscripts, conducted additional experiments, provided additional evidence and data, and significantly improved the quality of the research. The rebuttal to reviewers was also well-prepared and fairly addressed my previous concerns. However, in my opinion, the revised article still could not yet meet the expectations of this journal tier:

1) In the current form of the article, two separate teams of scientific contributions could be observed: (A) one from a team of data analysts and machine learning experts, who contributed computational methodologies, model developments/validations and discussions, and analytics codes; and (B) one from a team of data generators, clinical immunologist(s) and annotators.

There is, however, a mismatch between what (A) wishes to achieve and what (B) could provide. An example is the experimental setup for intra- and inter-rater variability testing: 100 labeled samples were selected to challenge 03 raters (one with an immunology background and two data analysts). Readers will not be able to appreciate the fidelity of the results based on such a small test set, with the inclusion of only one biologist.

We thank the reviewer for the time spent assessing our manuscript and for the positive remarks on our revision. We would like to emphasize that the machine learning experts and immunologists collaborated closely on this project for over three years and supported each other in every step of the work. The panel optimization and the selection of the stainings, experimental design, quality control of the experiments, comparison of the existing gating strategies vs. machine learning, donor selection, comparison of the experiments, and interpretation of the analysis results were conducted in close collaboration.

To address the very important point of the fidelity of the annotations, we performed an additional intra- and inter-rater variability analysis, now involving two biologists, and added the following paragraph to the results section:

Two donors were randomly selected to assess intra- and inter-rater variability within two experiments (donor 1 in experiment III and donor 7 in experiment IV). Next, 100 annotated samples from donor 1 and 224 annotated samples from donor 7 were randomly selected for reannotation. Four annotators with diverse backgrounds annotated the images (two immunology experts, including the original annotator and a new expert, one data analyst, and one IFC analyst). For donor 1, Cohen's kappa²³ scores were 0.84 (intra-), 0.80 (inter-), 0.79 (inter-), and 0.66 (inter-rater), respectively, compared to the original annotation by rater 1. For donor 7, comparing the original annotation with the new annotations yielded Cohen's kappa scores of 0.95 (intra-), 0.86

(inter-), 0.78 (inter-), and 0.75 (inter-rater), respectively. The strong agreement between the annotators, especially between the immunologists (scores > 0.8 are considered as almost perfect agreement ²³), gave us confidence in the reproducibility and validity of the original annotation (Supplementary Fig. 2f).

We also added a supplementary figure:

f

Supplementary Fig. 2f Intra- and inter-rater comparison of the annotation. Images from two donors were randomly selected and reannotated by four annotators to assess the quality of the original annotations (ground truth). Rater 1 and 2 are expert immunologists, and Rater 1 is the original annotator of the ground truth. There is a very strong agreement (Cohen kappa score ≥ 0.80) between expert immunologists.

2) The added text for explainable AI was fairly helpful, including Singh et al. (29) and Tjoa et al. (30). On the positive side, the authors showed attempts to deliver: (A) the list of human-readable feature names, (B) a model that could natively provide feature importance measure, and (C) biological meaning of the features. However, it is not yet sufficient to claim this explainability is novel in the field and better than state-of-the-art. More direct comparison, benchmark, or reference to truly state-of-the-art studies in 2019-2023 is needed for such a claim. Secondly, most of these explainable imagery features are presented through the lens of Amnis

ImageStreamX Mark II Imaging Flow cytometer, an instrument that had its first debut around ~2010s (Imaging flow cytometry technology itself is even older).

We would like to thank the reviewer for their remark. The observation that explainable AI (xAI) and imaging flow cytometer (IFC) have been around for some time is correct. In our work, the novelty comes from the fact that these two have not been used together previously. In particular, our work is the first coherent functional study using large IFC data sets and the consequential analytical part to visualize, understand and study synapse formation and its link to predictive features using IFC and xAI.

Nonetheless, we seized the opportunity to implement the reviewer's comment and improved the explainability paragraph of the manuscript. First, we elaborated in more detail about the interpretable features:

In order to characterize the immunological synapse in an unbiased fashion, we first designed and computed a series of biologically motivated, interpretable features using the scifAI framework. These features were based on morphology, intensity, co-localization, texture, and synaptic features extracted from the 5-panel stained images and their corresponding masks (Methods and Supplementary Fig. 3a-c). Morphology features included shape-related characteristics such as area, perimeter, and roundness of cells. Intensity and texture features were implemented to quantify the existence, distribution, and regularity of intensities within cells¹. Co-localization features were designed to capture similarities and differences in intensities among different channels². Finally, synaptic features were implemented to give insights into the distribution of intensities within the synapse region compared to the rest of the cell^{3,4}. Synaptic features were implemented based on the signal intensity ratio of each fluorescent channel in the synaptic area to the whole cell. These features allowed for comparing cell states within and among different populations.

The aforementioned features can be used in other fluorescent microscopy datasets and are not exclusive to IFC.

Moreover, we rewrote the explainability paragraph, including more details about the intrinsic mechanism used. In addition, we added a new state-of-the-art explainability method, SHAP, as a sanity check, as well as the corresponding figures and the reference to a recent study that shows SHAP is the most used method in related topics:

Next, we focused on the explainability of the pipeline and explored which underlying features drive the class prediction, ranking features by their respective feature importance. The feature importance was based on the

in-model mechanism called gain⁵, which signifies the relative contribution of a corresponding feature to the classification (Methods). The most predictive features were based on the intensity of CD3, co-localization of MHCII & P-CD3ζ, co-localization of MHCII & CD3, the cell morphology in BF, and the intensity of the MHCII (Fig. 1d). The intensities of CD3 and MHCII imply the presence of a T cell or B-LCL in the image. The co-localization of MHCII and P-CD3ζ measures how many overlapping pixels the two proteins share. For example, `synapse w/ signaling` has a lower overlap between the T cell and the B-LCL than the `T cell & B-LCL in one layer,` and thus a lower co-localization of MHCII and P-CD3ζ. The same logic can be applied to the co-localization of MHCII and CD3. The equivalent diameter of BF hints at the size of the cells or the existence of multiple cells. Based on the features and the definition of classes, one could speculate that the classifier uses (i) the intensity of CD3 and MHCII to detect the existence of T and B-LCL cells in the image, (ii) the co-localization of MHCII & P-CD3ζ as well as MHCII & CD3 to detect the different doublets types and the existence of signaling (iii) the cell morphology in BF to assist detecting the cell type (Fig. 1d). To validate the feature importance with a widely used, post-model explainability approach, SHAP⁶ values were calculated (Methods). SHAP is a game theory-based approach that has recently been shown to be the most used post-model explainability method in related topics⁷. The top-5 SHAP values (Supplementary Fig. 5c) were similar to the previously shown feature importances based on model-intrinsic gain, providing additional confidence in the stability of the feature importance estimation (Fig. 1d).

3) There was more than one reviewer who noticed the generalizability issue of the model. The authors professionally conducted several validation settings for the XGBoost and ResNet. However, it is questionable why only XGBoost could be this good for XAI, in general, and in this research in particular.

We would like to emphasize that large performance gains in prediction are often achieved by a high-quality feature space, learned intrinsically, e.g., by CNNs on large data sets, or by feature engineering, including prior knowledge, such as in our approach. Therefore, we performed an ablation study to show that the quality of the interpretable features is the most important factor in our model's performance and added the following paragraph to the Result section:

An interesting observation was that even though the autoencoder has been trained on all unlabeled data and thus was allowed to learn a data-driven representation of the full data set, this derived feature space is considerably less performant as compared to the engineered interpretable features in a logistic regression model (0.79 ± 0.02 vs. 0.90 ± 0.02 , Fig. 1c). To confirm the quality of the interpretable features, the previously trained XGBoost and

logistic regression models were compared with different classifiers in an ablation study on the interpretable features. These classifiers included random forest, support vector machine, and linear discriminant analysis. While XGBoost performed best (0.92 ± 0.02), we only observed minor drops in performance using random forest (RF, 0.90 ± 0.02), linear regression (LR, 0.90 ± 0.02), support vector machine (SVM, 0.90 ± 0.02), or linear discriminant analysis (LDA, 0.87 ± 0.02). Thus, we concluded that the predictive performance driving factor is the feature space of interpretable features (Supplementary Fig. 5a).

a

Supplementary Figure 5a: Ablation study: comparison of additional classifiers on the interpretable features, including XGboost, random forest (RF), logistic regression (LR), support vector machine (SVM), and linear discriminant analysis (LDA). While XGBoost exhibits the highest performance, other classifiers with different capacities show very good performance, which suggests that interpretable features are the driving factor for good performance.

In a nutshell, we demonstrated that the choice of classifier had a small impact. It is clear that there is no considerable difference among the classifiers with different capacities, and we attribute this to the quality of the interpretable features.

Reviewer #3 (Remarks to the Author):

The authors provide satisfying responses and revisions of the paper in response to the reviewers' comments.

We thank the reviewer for previous suggestions and for acknowledging the importance of our work.

Reviewer #4 (Remarks to the Author):

The authors have provided helpful additional explanation of the meaning of explainable AI vs interpretation and other points of common concern to the reviewers. The discussion of how generalisable this dataset could be perhaps gives up on this too easily. As far as I'm aware there is really one instrument that does imaging flow cytometry. The staining reagents that are being used are also very standard commercial reagents. Could the author envision a pipeline to calibrate imaging conditions to guide a user to adopt conditions that would allow a human peripheral blood T cell to B-LCL conjugate system to be brought into range with the parameters to the trained model as downloaded from GitHub? This might be similar to a bead calibration to generate consistent results in flow cytometry, but a step beyond this to bring all staining conditions and machine setting into a range where the AI could correctly segment the cells and synapse. Perhaps this could be pursued with Amnis as this reverse training could also be helpful to standardize machines and generally support reproducibility in a community. I don't expect you to address here, but it might be worth considering this in the future. Otherwise I have no further concerns.

We thank the reviewer for previous suggestions and for acknowledging the importance of our work. We added a new paragraph to shed more light on the generalizability of the model. Regarding the IFC calibration for similar applications, we absolutely agree with the reviewer's remark and thank them.

References

1. Caicedo, J. C. *et al.* Data-analysis strategies for image-based cell profiling. *Nat Methods* 14, 849–863 (2017).
2. Dunn, K. W., Kamocka, M. M. & McDonald, J. H. A practical guide to evaluating colocalization in biological microscopy. *Am J Physiol-cell Ph* 300, C723–C742 (2011).
3. Wabnitz, G., Kirchgessner, H. & Samstag, Y. Qualitative and Quantitative Analysis of the Immune Synapse in the Human System Using Imaging Flow Cytometry. *J Vis Exp* (2019) doi:10.3791/55345.
4. Markey, K. A., Gartlan, K. H., Kuns, R. D., MacDonald, K. P. A. & Hill, G. R. Imaging the immunological synapse between dendritic cells and T cells. *J Immunol Methods* 423, 40–44 (2015).
5. Krishnapuram, B. *et al.* XGBoost. *Proc. 22nd ACM SIGKDD Int. Conf. Knowl. Discov. Data Min.* 785–794 (2016) doi:10.1145/2939672.2939785.
6. Lundberg, S. M. *et al.* From local explanations to global understanding with explainable AI for trees. *Nat Mach Intell* 2, 56–67 (2020).
7. Loh, H. W. *et al.* Application of explainable artificial intelligence for healthcare: A systematic review of the last decade (2011–2022). *Comput. Methods Programs Biomed.* 226, 107161 (2022).
8. German, Y. *et al.* Morphological profiling of human T and NK lymphocytes by high-content cell imaging. *Cell Reports* 36, 109318 (2021).
9. Xiong, W. *et al.* Immunological Synapse Predicts Effectiveness of Chimeric Antigen Receptor Cells. *Mol Ther* 26, 963–975 (2018).
10. Naghizadeh, A. *et al.* In vitro machine learning-based CAR T immunological synapse quality measurements correlate with patient clinical outcomes. *Plos Comput Biol* 18, e1009883 (2022).
11. Rees, P., Summers, H. D., Filby, A., Carpenter, A. E. & Doan, M. Imaging flow cytometry. *Nat. Rev. Methods Primers* 2, 86 (2022).
12. Ren, P. *et al.* A Survey of Deep Active Learning. *Acm Comput Surv* 54, 1–40 (2021).
13. Boushehri, S. S., Qasim, A., Waibel, D., Schmich, F. & Marr, C. Systematic comparison of incomplete-supervision approaches for biomedical imaging classification. (2021) doi:10.21203/rs.3.rs-798207/v1.
14. Ahmed, F., Friend, S., George, T. C., Barteneva, N. & Lieberman, J. Numbers matter: Quantitative and dynamic analysis of the formation of an immunological synapse using imaging flow cytometry. *J Immunol Methods* 347, 79–86 (2009).
15. Saeys, Y., Gassen, S. V. & Lambrecht, B. N. Computational flow cytometry: helping to make sense of high-dimensional immunology data. *Nat. Rev. Immunol.* 16, 449–462 (2016).
16. Kranich, J. *et al.* In vivo identification of apoptotic and extracellular vesicle-bound live cells using image-based deep learning. *J Extracell Vesicles* 9, 1792683 (2020).
17. Hu, Z., Bhattacharya, S. & Butte, A. J. Application of Machine Learning for Cytometry Data. *Front. Immunol.* 12, 787574 (2022).

18. Camargo, G., Bugatti, P. H. & Saito, P. T. M. Active semi-supervised learning for biological data classification. *PLoS ONE* 15, e0237428 (2020).
19. Zbontar, J., Jing, L., Misra, I., LeCun, Y. & Deny, S. Barlow Twins: Self-Supervised Learning via Redundancy Reduction. *Arxiv* (2021) doi:10.48550/arxiv.2103.03230.
20. Lippeveld, M. et al. Classification of Human White Blood Cells Using Machine Learning for Stain-Free Imaging Flow Cytometry. *Cytom Part A* 97, 308–319 (2020).
21. Eulenberg, P. et al. Reconstructing cell cycle and disease progression using deep learning. *Biorxiv* 081364 (2017) doi:10.1101/081364.
22. Zhang, Z.-Y., Zhao, P., Jiang, Y. & Zhou, Z.-H. Learning From Incomplete and Inaccurate Supervision. *Ieee T Knowl Data En* 34, 5854–5868 (2022).
23. McHugh, M. L. Interrater reliability: the kappa statistic. *Biochem Medica* 22, 276–282 (2012).

Reviewers' Comments:

Reviewer #1:

Remarks to the Author:

The author has carefully addressed the comments and made revisions to the manuscript concerning concerns related to the dataset and feature selection. However, there are still a few remaining issues that need to be addressed:

1. In the section "scifAI enables high-throughput profiling of the immunological synapse," the dataset is stated to contain 5221 annotated images. However, all experiments were conducted on the \pm SEA-based subset, which only included 4490 images. It is unclear why the complete dataset of 5221 images was not utilized. Please explain this discrepancy.
2. The dataset was collected from nine donors, consisting of two women and seven men. It is important to clarify whether the author considered gender when dividing the dataset or if any gender-related analysis was conducted.
3. The main text mentions: "Moreover, the framework implements several machine learning and deep learning models for training supervised image classification models, e.g., for the prediction of cell configurations such as the immunological synapse." Please specify the specific model(s) used to avoid ambiguity in the expressions.
4. The main text mentions: "In order to characterize the immunological synapse in an unbiased fashion... These features were based on morphology, intensity, co-localization, texture, and synaptic features extracted from the 5-panel stained images and their corresponding masks." It is important to note that the characterization of the immunological synapse is claimed to be done in an unbiased fashion. However, using a small test sample of 5-panel-stained images and their corresponding masks may lack sufficient support for this claim. Please address this concern and provide further justification or discuss potential limitations.
5. In the section on "Interpretable feature engineering from images" in the Materials & Methods, the author lists 296 biologically motivated features for studying the immunological synapse. These features include morphology, intensity, co-localization, texture, and synaptic-related values. This appears to be a complex method. How were these 296 features jointly analyzed, and is there any relevant theory or methodology to support this approach? Please provide further details and clarify the analysis process.
6. There are some format errors throughout the manuscript. Subtitles such as "Interpretable feature engineering from images," "Autoencoder feature extraction," "Feature pre-selection," "Classical supervised learning models," and "Convolutional neural networks" should be bolded for consistency and clarity. Please ensure that these subtitles are properly formatted.

Reviewer #2:

Remarks to the Author:

In this round of revision, the authors have demonstrated additional efforts in

(A) addressing lingering concerns in generalizability by additional intra- and inter-rater variability analysis (now involving two more biologists, Supplementary Fig. 2f)

(B) rewriting explainability, adding important details and analyses (including ablation study, Supplementary Fig. 5a)

(C) presenting both model-centric and data-centric evaluations (quantitatively comparing several models XGBoost, SVM, Random Forest, CNNs etc.; while leveraging both annotated and unlabelled data)

Compared to their first version of the manuscript, the authors have shown genuine and constructive collaborations with the reviewers to improve the merits of the work. I recommend a favorable consideration to publish this work, which would encourage the community to focus more on explainable AI applications in biomedical studies.

Minor suggestions:

- 1) Please revise and rewrite significantly figure captions for both main and supplementary materials. Most supplementary figures currently contain a 1- or 2-sentence caption per panel of the figure. Take Supplementary Figure 1 for example, while captions for panels a-c are succinct for experts in the cytometry field, the general audience would gain little information reading these short captions. Another example: please find a more data-science substitution for "...very good performance..." in Supplementary Figure 5a.
- 2) Since intensity features are of high importance in this work, please include expand the details of the compensation matrix and if possible, include the used matrices as supplementary materials.
- 3) Since "Gain" of XGBoost was important to determine in-model feature relevance, please expand its discussion (currently just a reference 43; the Method section is also lacking details of its use and formulas).
- 4) The GitHub repo for this work is neatly organized. Please continue to improve its documentation and instructions for the general audience.

REVIEWER COMMENTS

Reviewer #1 (Remarks to the Author):

The author has carefully addressed the comments and made revisions to the manuscript concerning concerns related to the dataset and feature selection. However, there are still a few remaining issues that need to be addressed:

1. In the section "scifAI enables high-throughput profiling of the immunological synapse," the dataset is stated to contain 5221 annotated images. However, all experiments were conducted on the \pm SEA-based subset, which only included 4490 images. It is unclear why the complete dataset of 5221 images was not utilized. Please explain this discrepancy.

We thank the reviewer for this point. The difference in the numbers is the consequence of the train/test split. We clarified this point by adding more information in the Results section:

To estimate the performance of the models, the annotated \pm SEA-based dataset with 5,221 images in total was split into train (70%) and test (30%) sets, resulting in 3,654 images for training and 1,567 images for testing. All models were trained on the stratified \pm SEA-based training set. We compared the macro F1-score on the \pm SEA-based hold-out test set to benchmark the classification model and feature space combinations as the (Methods).

We also elaborated on this in the Methods section:

In addition, our expert annotated a subset of data for -SEA (labeled images=1,160), +SEA (4,061), CD19-TCB (396), and Teplizumab (227). The labeled \pm SEA-based data (5,221) was used for training and validation of the classification models, where 70% was used for training and 30% for testing in a stratified way.

2. The dataset was collected from nine donors, consisting of two women and seven men. It is important to clarify whether the author considered gender when dividing the dataset or if any gender-related analysis was conducted.

Thank you for raising this important issue. Based on the limited number of donors in general and the data set comprising only two female donors in particular, we refrained from conducting a specific analysis on this point as it would not have been statistically reliable. Nevertheless, we included the gender of the donors in Supplementary Figure 2 to ensure transparency.

3. The main text mentions: "Moreover, the framework implements several machine learning and deep learning models for training supervised image classification models, e.g., for the prediction of cell configurations such as the immunological synapse." Please specify the specific model(s) used to avoid ambiguity in the expressions.

We revised the manuscript in the Results section to address this concern:

Moreover, the framework implements several machine learning and deep learning models for training supervised image classification models, e.g., for predicting cell configurations such as the immunological synapse. The implemented models included logistic regression ¹, linear discriminant analysis ¹, support vector machine ¹, random forest ², and XGBoost ³, as well as deep learning models including a multi-encoder autoencoder ⁴, DeepFlow ⁵, ResNet18 and ResNet34 ⁶.

Detailed model names are also provided in the performance evaluation, as shown in Figure 1c and Supplementary Figure 6a.

4. The main text mentions: "In order to characterize the immunological synapse in an unbiased fashion... These features were based on morphology, intensity, co-localization, texture, and synaptic features extracted from the 5-panel stained images and their corresponding masks." It is important to note that the characterization of the immunological synapse is claimed to be done in an unbiased fashion. However, using a small test sample of 5-panel-stained images and their corresponding masks may lack sufficient support for this claim. Please address this concern and provide further justification or discuss potential limitations.

We thank the reviewer for raising this point. The stainings were carefully selected to reflect the cell structure and capture a wide range of biologically motivated, potentially relevant immunological synapse characteristics. For example, the enrichment of F-actin or P-CD3 ζ in the synaptic area shows the formation of immunological synapses. While we demonstrated that careful selection of a relevant set of markers enabled us to identify relevant characteristics of the immunological synapse, it can potentially add biases to the analysis. Nonetheless, "scifAI" is designed to be unbiased towards any IFC dataset with an arbitrary set of stainings. The term *unbiased* refers to how scifAI supports feature engineering based on a given set of markers, as defined in the experimental design.

We seize the opportunity to add more information about the choice of stainings and appropriate references in the Results section, where we now write:

The designed multi-channel panel consisted of brightfield (BF), F-actin (cytoskeleton), MHCII, CD3, and P-CD3 ζ (TCR signaling), allowing to capture a wide range of biologically motivated, potentially relevant characteristics of the immunological synapse (Fig. 1a) ^{7,8}.

Moreover, we added the potential limitation in the Discussion section:

The combination of interpretable features and explainable machine learning enabled us to identify various relevant classes, such as immunological synapses, with state-of-the-art accuracy. It also allowed us to investigate the morphological profiles of the immunological synapse in an unbiased way and characterize the mode of action of antibodies in a biologically relevant context. This methodology is thus a substantial contribution to the field, which has so far primarily focused on performance over interpretability by using a ResNet CNN architecture as the backbone ^{9,10}. A potential limitation in this work is the choice of markers. While the markers were carefully selected to reflect immunological synapse characteristics, our results are restricted to those choices. Nonetheless, feature engineering in scifAI is designed in an unbiased way towards any IFC dataset with an arbitrary set of stainings. As proof of concept, we provide three examples within the scifAI code repository on how to analyze IFC datasets from Jurkat cells (3 channels per image) ¹¹, white blood cells (12 channels) ¹², and apoptotic cells (2 channels) ⁴ using scifAI.

5. In the section on "Interpretable feature engineering from images" in the Materials & Methods, the author lists 296 biologically motivated features for studying the immunological synapse. These features include morphology, intensity, co-localization, texture, and synaptic-related values. This appears to be a complex method. How were these 296 features jointly analyzed, and is there any relevant theory or methodology to support this approach? Please provide further details and clarify the analysis process.

It is very common in computer vision to extract features from images instead of directly using pixel values¹³. The goal is to reduce the dimensional space and focus on relevant information in images. The images in our study had, on average, a size of 5x80x80 (number of channels x height x width). Using the pixels directly would lead to a 32,000-dimensional feature space, which most machine learning algorithms will fail in finding patterns due to the curse of dimensionality¹. Instead, finding and extracting features from images is desirable to reduce the dimensionality of the problem and to handle meaningful, interpretable features. An alternative choice to define features is to use deep learning techniques to learn a useful yet abstract representation of the images.

In our work, we focused on a set of features that are meaningful based on the biology of immunological synapses, inspired by previous works^{4,11,14-20}. We analyzed these features in a variety of scenarios:

- For classifying synapses, all the features were used in a joint manner in a "feature pre-selection pipeline" based on the work of Haq et al.²¹ and then used an XGBoost³ model, which again uses all the features (Fig. 1c and Supplementary Fig. 5).
- For understanding the mode-of-action of antibodies, different conditions were compared based on these features using statistical tests, which is common practice in single-cell profiling¹⁷. All the statistical tests were also corrected using multiple testing corrections to ensure the sanity of the results (Fig 2&3).
- Again, these features were used together to predict cytokines using regression analysis, which is standard practice for predicting continuous values using multi-dimensional data²² (Fig. 4).

6. There are some format errors throughout the manuscript. Subtitles such as "Interpretable feature engineering from images," "Autoencoder feature extraction," "Feature pre-selection," "Classical supervised learning models," and "Convolutional neural networks" should be bolded for consistency and clarity. Please ensure that these subtitles are properly formatted.

Thanks for pointing out the format errors. We corrected them in the latest version of our manuscript.

Reviewer #2 (Remarks to the Author):

In this round of revision, the authors have demonstrated additional efforts in

(A) addressing lingering concerns in generalizability by additional intra- and inter-rater variability analysis (now involving two more biologists, Supplementary Fig. 2f)

(B) rewriting explainability, adding important details and analyses (including ablation study, Supplementary Fig. 5a)

(C) presenting both model-centric and data-centric evaluations (quantitatively comparing several models XGBoost, SVM, Random Forest, CNNs etc.; while leveraging both annotated and unlabelled data)

Compared to their first version of the manuscript, the authors have shown genuine and constructive collaborations with the reviewers to improve the merits of the work. I recommend a favorable consideration to publish this work, which would encourage the community to focus more on explainable AI applications in biomedical studies.

We thank the reviewer for seeing value in our work.

Minor suggestions:

1) Please revise and rewrite significantly figure captions for both main and supplementary materials. Most supplementary figures currently contain a 1- or 2-sentence caption per panel of the figure. Take Supplementary Figure 1 for example, while captions for panels a-c are succinct for experts in the cytometry field, the general audience would gain little information reading these short captions. Another example: please find a more data-science substitution for "...very good performance..." in Supplementary Figure 5a.

We thank the reviewer for raising this point. We revisited both supplementary figure captions and expanded explanations where appropriate.

2) Since intensity features are of high importance in this work, please include expand the details of the compensation matrix and, if possible, include the used matrices as supplementary materials.

We have added the following text in the Methods section and now provide all compensation matrices in Supplementary Figure 2.

Immune synapse formation and imaging flow cytometry

[...] Only synapses that showed a CD3 signal in the mask were gated. Finally, T+B-LCL cells in one layer were excluded using the brightfield (BF) height and area feature, and single T-B-LCL synapses were analyzed. For each experiment, a compensation matrix was calculated to minimize spillovers into the different channels (see Supplementary Fig. 2).

Preparation of the imaging dataset for analysis

[...] The images contained brightfield (BF), F-actin, MHCII, CD3, P-CD3 ζ , and Live/Dead stainings. The Live/Dead staining is only used to filter out the dead cells. For each experiment, the images were compensated using a compensation matrix derived from stained single cells. After the compensation, the raw images (16-bit) and their corresponding channel-wise segmentation masks were exported from the IDEAS software and saved in an HDF5 format.

Supplementary Figure 2: Compensation matrices to minimize spillovers into the different channels for all experiments For each experiment, a new compensation matrix was generated. Subfigures a-d represent experiments I-IV, respectively.

3) Since "Gain" of XGBoost was important to determine in-model feature relevance, please expand its discussion (currently just a reference 43; the Method section is also lacking details of its use and formulas).

We thank the reviewer for bringing up this point. We have added the following paragraph to the Methods section:

Feature importance of XGBoost model

Average gain was used to determine the XGBoost model's feature importance. To calculate the average gain for a specific feature, it is necessary to find all trees and splits which used that feature. Then, the improvement by each split in the multi-class loss function is calculated. Finally, the average gain is calculated by summing all improvements and dividing this sum by the total number of splits involving the feature of interest.

4) The GitHub repo for this work is neatly organized. Please continue to improve its documentation and instructions for the general audience.

We thank the reviewer for bringing this point up. We will maintain the repository and keep it updated.

References

1. Hastie, T., Tibshirani, R. & Friedman, J. The Elements of Statistical Learning, Data Mining, Inference, and Prediction. *Springer Ser. Stat.* (2009) doi:10.1007/978-0-387-84858-7.
2. Breiman, L. Random Forests. *Mach. Learn.* 45, 5–32 (2001).
3. Krishnapuram, B. *et al.* XGBoost. *Proc. 22nd ACM SIGKDD Int. Conf. Knowl. Discov. Data Min.* 785–794 (2016) doi:10.1145/2939672.2939785.
4. Kranich, J. *et al.* In vivo identification of apoptotic and extracellular vesicle-bound live cells using image-based deep learning. *J Extracell Vesicles* 9, 1792683 (2020).
5. Eulenberg, P. *et al.* Reconstructing cell cycle and disease progression using deep learning. *Nat Commun* 8, 463 (2017).
6. He, K., Zhang, X., Ren, S. & Sun, J. Deep Residual Learning for Image Recognition. *2016 IEEE Conf. Comput. Vis. Pattern Recognit. (CVPR)* 770–778 (2016) doi:10.1109/cvpr.2016.90.
7. Wabnitz, G. H., Nessmann, A., Kirchgessner, H. & Samstag, Y. InFlow microscopy of human leukocytes: A tool for quantitative analysis of actin rearrangements in the immune synapse. *J Immunol Methods* 423, 29–39 (2015).
8. Schubert, D. A. *et al.* Self-reactive human CD4 T cell clones form unusual immunological synapses. *J Exp Medicine* 209, 335–352 (2012).
9. Perakis, A. *et al.* Contrastive Learning of Single-Cell Phenotypic Representations for Treatment Classification. *Arxiv* (2021) doi:10.1007/978-3-030-87589-3_58.
10. Doan, M. *et al.* Deepometry, a framework for applying supervised and weakly supervised deep learning to imaging cytometry. *Nat Protoc* 16, 3572–3595 (2021).
11. Eulenberg, P. *et al.* Reconstructing cell cycle and disease progression using deep learning. *Biorxiv* 081364 (2017) doi:10.1101/081364.
12. Lippeveld, M. *et al.* Classification of Human White Blood Cells Using Machine Learning for Stain-Free Imaging Flow Cytometry. *Cytom Part A* 97, 308–319 (2020).
13. Mahadevkar, S. V. *et al.* A Review on Machine Learning Styles in Computer Vision—Techniques and Future Directions. *IEEE Access* 10, 107293–107329 (2022).
14. Ahmed, F., Friend, S., George, T. C., Barteneva, N. & Lieberman, J. Numbers matter: Quantitative and dynamic analysis of the formation of an immunological synapse using imaging flow cytometry. *J Immunol Methods* 347, 79–86 (2009).
15. Blasi, T. *et al.* Label-free cell cycle analysis for high-throughput imaging flow cytometry. *Nat Commun* 7, 10256 (2016).
16. Carpenter, A. E. *et al.* CellProfiler: image analysis software for identifying and quantifying cell phenotypes. *Genome Biol* 7, R100–R100 (2006).
17. Heumos, L. *et al.* Best practices for single-cell analysis across modalities. *Nat. Rev. Genet.* 1–23 (2023) doi:10.1038/s41576-023-00586-w.
18. Hennig, H. *et al.* An open-source solution for advanced imaging flow cytometry data analysis using machine learning. *Methods San Diego Calif* 112, 201–210 (2017).
19. Dunn, K. W., Kamocka, M. M. & McDonald, J. H. A practical guide to evaluating colocalization in biological microscopy. *Am J Physiol-cell Ph* 300, C723–C742 (2011).
20. German, Y. *et al.* Morphological profiling of human T and NK lymphocytes by high-content cell imaging. *Cell Reports* 36, 109318 (2021).
21. Haq, A. U., Zhang, D., Peng, H. & Rahman, S. U. Combining Multiple Feature-Ranking Techniques and Clustering of Variables for Feature Selection. *Ieee Access* 7, 151482–151492 (2019).
22. Gupta, A., Sharma, A., Goel, D. A. & University, M. A. I. of T. G. S. I. Review of Regression Analysis Models. *Int. J. Eng. Res.* V6, (2017).